# EB1 decoration of microtubule lattice facilitates spindle-kinetochore lateral attachment in *Plasmodium* male gametogenesis

Shuzhen Yang[1,4], Mengya Cai [1,4], Junjie Huang[2,4], Shengnan Zhang[1], Xiaoli Mo[1], Kai Jiang [2,3] ✉, Huiting Cui[1] ✉ & Jing Yuan [1] ✉

Faithful chromosome segregation of 8 duplicated haploid genomes into 8 daughter gametes is essential for male gametogenesis and mosquito transmission of *Plasmodium*. *Plasmodium* undergoes endomitosis in this multinucleated cell division, which is highly reliant on proper spindle-kinetochore attachment. However, the mechanisms underlying the spindle-kinetochore attachment remain elusive. End-binding proteins (EBs) are conserved microtubule (MT) plus-end binding proteins and play an important role in regulating MT plus-end dynamics. Here, we report that the *Plasmodium* EB1 is an orthologue distinct from the canonical eukaryotic EB1. Both in vitro and in vivo assays reveal that the *Plasmodium* EB1 losses MT plus-end tracking but possesses MT-lattice affinity. This MT-binding feature of *Plasmodium* EB1 is contributed by both CH domain and linker region. EB1-deficient parasites produce male gametocytes that develop to the anucleated male gametes, leading to defective mosquito transmission. EB1 is localized at the nucleoplasm of male gametocytes. During the gametogenesis, EB1 decorates the full-length of spindle MTs and regulates spindle structure. The kinetochores attach to spindle MTs laterally throughout endomitosis and this attachment is EB1-dependent. Consequently, impaired spindle-kinetochore attachment is observed in EB1-deficient parasites. These results indicate that a parasite-specific EB1 with MT-lattice binding affinity fulfills the spindle-kinetochore lateral attachment in male gametogenesis.

Malaria, caused by the protozoan parasites of the genus *Plasmodium*, is an infectious disease causing 241 million cases and 620 thousand deaths in 2020[1]. The parasitism of *Plasmodium* switches between vertebrate hosts and female *Anopheles* mosquito vectors. In the vertebrate hosts, the *Plasmodium* undergoes asexual multiplication in the hepatocytes and then in the erythrocytes. Sexual development starts with a small proportion of intra-erythrocyte parasites irreversibly differentiating into female and male gametocytes, the sexual

[1]State Key Laboratory of Cellular Stress Biology, School of Life Sciences, Faculty of Medicine and Life Sciences, Xiamen University, 361102 Xiamen, Fujian, China. [2]The State Key Laboratory Breeding Base of Basic Science of Stomatology and Key Laboratory of Oral Biomedicine Ministry of Education, School & Hospital of Stomatology, Medical Research Institute, Wuhan University, Wuhan, China. [3]Frontier Science Center for Immunology and Metabolism, Wuhan University, Wuhan, China. [4]These authors contributed equally: Shuzhen Yang, Mengya Cai, Junjie Huang. ✉e-mail: jiangkai@whu.edu.cn; cuihuiting@xmu.edu.cn; yuanjing@xmu.edu.cn

precursors essential for mosquito transmission[2]. After ingestion by mosquitoes, gametocytes are activated by midgut factors, differentiate into gametes and egress from the erythrocytes, a process known as gametogenesis[3]. While a female gametocyte produces a female gamete, a male gametocyte gives rise to 8 male gametes[4]. Male and female gametes fertilize and develop into ookinete, which traverses the mosquito midgut and transforms to oocyst, and each oocyst produces thousands of sporozoites[5]. When mosquitoes bite again, the sporozoites in the salivary glands are injected into a new vertebrate host.

During its life cycle, the *Plasmodium* does not proliferate by binary fission. Instead, it employs diverse multinucleated modes including the hepatic and erythrocytic schizogony in vertebrates as well as the male gametogenesis and oocyst sporogony in mosquitoes[6]. Among them, male gametogenesis is an extraordinarily rapid process. Genome duplication and chromosome segregation, producing 8 haploid male gametes from a single progenitor, occurs within 10–12 min[7,8]. Two spatially distinct components are coordinated during this process. One is the assembly of 8 axonemes in the cytosol. The other includes the three rounds of DNA replication and endomitosis (I, II, and III) without nuclear division, resulting in an octoploid nucleus with 8 hemispindles. A bipartite microtubule (MT) organization center (MTOC)[9], which associates with the nuclear envelope, orchestrates the assembly of two distinct MT organizations spindle and axoneme. Within MTOC, the nuclear spindle pole serves as the nucleation template for spindle while the cytoplasmic basal body for axoneme. Subsequent exflagellation via the attached axonemes results in 8 haploid male gametes egressing from the parasite body of male gametocytes.

Mitotic spindle is the MT cytoskeleton responsible for chromosome segregation. *Plasmodium* endomitosis differs remarkably from the canonical mitosis in which the centrosome-derived bipolar spindles connect with kinetochores via MT-kinetochore end-on attachment using the search-and-capture model[10,11]. In *Plasmodium* male gametogenesis, first hemispindle assembles from the spindle pole at the beginning of endomitosis I and finally 8 hemispindles form by an unknown mechanism by the end of endomitosis III[9,12]. In addition, the chromosomes are not condensed and not segregated until the end of endomitosis III[13,14], when one hemispindle presumably captures one set of kinetochores of haploid chromosomes. It remains unclear how the *Plasmodium* achieves correct pairing and attachment between hemispindles and kinetochores in this fast and continuous multinucleated endomitosis. Cell imaging of fluorescently tagged kinetochore proteins NDC80 has revealed that kinetochores undergo three rounds of duplication and separation as well[15,16]. These results suggest that the first hemispindle establishes attachment with the kinetochores from endomitosis I and this attachment maintains throughout the endomitosis for coordinated duplication of spindle, kinetochore, and chromosome. Elucidation of the *Plasmodium* endomitotic spindle has long been hampered due to the limited resolution via conventional electron or fluorescence microscopy[15–19]. Nevertheless, the fine structure of hemispindles, the spatiotemporal relationship between spindle and kinetochores, and the mechanism for spindle-kinetochore attachment during male gametogenesis, remain largely unknown.

End-binding proteins (EBs) are a class of MT plus-end tracking proteins that function as a hub for protein interaction at the growing MT plus-end and regulate MT dynamics[20,21]. EB harbors an N-terminal calponin homology (CH) domain, a C-terminal coiled-coil (CC) domain, and an evolutionarily divergent linker[22]. Biochemical and structural reconstitutions had revealed that EBs utilize the CH domain to autonomously bind to the growing MT plus-end and the binding affinity of EB1 towards MT plus-end is 10-fold higher compared to the MT lattice[23–26]. EBs are evolutionarily conserved in eukaryotes, from yeast to human[27–30]. Only one EB protein, EB1, is found in the *Plasmodium*, the precise localization and function of which in the life cycle of *Plasmodium* remains largely unexplored.

In this study, we report that the *Plasmodium* EB1 is an orthologue distinct from the canonical EB1. Both in vitro and in vivo assays reveal that the *Plasmodium* EB1 lost the MT plus-end tracking but has the MT-lattice binding affinity. In addition, we observe that kinetochores connect with the spindle at the MT lateral side, an unusual phenomenon different from the MT-kinetochore end-on attachment. Genetic disruption of EB1 leads to impaired spindle-kinetochore attachment for chromosome segregation in the male gametogenesis. These findings support a structural and functional link between the MT-lattice binding of EB1 and the spindle MT-kinetochore lateral attachment.

## Results

### EB1 is expressed at the nucleoplasm in the gametocytes

To define the expression pattern of EB1 in the parasite's life cycle, we tagged the *eb1* (PY17X_0407900) with a sextuple HA (6HA) epitope at the carboxyl (C)-terminus in the *P. yoelii* 17XNL strain (wildtype, WT) using the CRISPR-Cas9 method[31,32]. Immunofluorescence assay (IFA) of the tagged strain *eb1::6HA* showed that EB1 was expressed at the sexual stage gametocytes but not at asexual blood stages (including the ring, trophozoite, and schizont) in mice (Fig. 1a). Immunoblot confirmed gametocyte-specific expression of EB1 (Fig. 1b). This expression pattern of EB1 in *P. yoelii* is consistent with previous transcriptome analyses which detected the *eb1* transcripts in gametocytes, but not in the asexual blood stages of *P. falciparum* and *P. berghei* (19, 20). In the parasite-infected mosquitoes, EB1 was detected in the midgut oocysts and salivary gland sporozoites (Fig. 1a). To dissect whether EB1 expression is gender-specific, the gametocytes were stained with the antibody recognizing both α-Tubulin I and α-Tubulin II proteins (a male gametocyte marker)[33] as well as anti-HA antibody. EB1 was present in both gender gametocytes with higher expression level in male (Fig. 1c). Notably, fluorescent signals of EB1 were observed in the nucleus of gametocytes (Fig. 1c), which was further confirmed by immunoblot of nuclear or cytoplasmic fractions from the gametocyte extracts (Fig. 1d).

### EB1 is essential for parasite transmission in mosquitoes

To investigate the function of EB1 in the parasite, we generated a mutant clone by deleting the entire coding sequence (1293 bp) of *eb1* in the 17XNL strain (Fig. 1e). The resulting mutant (referred to as *Δeb1*) displayed normal asexual blood stage proliferation and gametocyte formation in mice (Fig. 1f, g). Therefore, EB1 is not required for parasite blood stage proliferation and gametocyte differentiation in mice. To evaluate the potential roles of EB1 in parasite development in the mosquito, parasite-infected mice were fed to the *Anopheles stephensi* mosquitoes. Midgut oocysts and salivary gland sporozoites were counted in the mosquitoes on day 7 and 14 post infection, respectively. The *Δeb1* parasite produced significantly less number of midgut oocysts (Fig. 1h) and no salivary gland sporozoites (Fig. 1i), indicating failure of parasite transmission in the mosquitoes. To confirm that the above phenotype is caused by EB1 depletion, we reintroduced a sequence consisting of the coding region of *P. yoelii* *eb1* fused with 6HA tag to the *eb1* locus in *Δeb1*, generating the complementation strain (referred to as *Comp1*) (Fig. 1e). EB1::6HA fusion protein was expressed in the nucleus of both male and female gametocytes (Fig. 1j, k). Importantly, the *Comp1* parasites produced midgut oocysts in the mosquitoes at a similar level as that of WT (Fig. 1l), confirming that the defects observed in *Δeb1* were due to *eb1* disruption. The animo acid sequences of EB1 proteins are highly similar among the *Plasmodium* species (Supplementary Fig. 1), suggesting conserved functions. In agree with this hypothesis, complementation of *Δeb1* with the *P. falciparum* *eb1* (*Pfeb1*) efficiently restored the midgut oocyst formation in the mosquitoes (see the complementation strain *Comp2* in Fig. 1e, j, l). Therefore, EB1 is important for the *Plasmodium* development in the mosquitoes and is functionally conserved in both *P. yoelii* and *P. falciparum*.

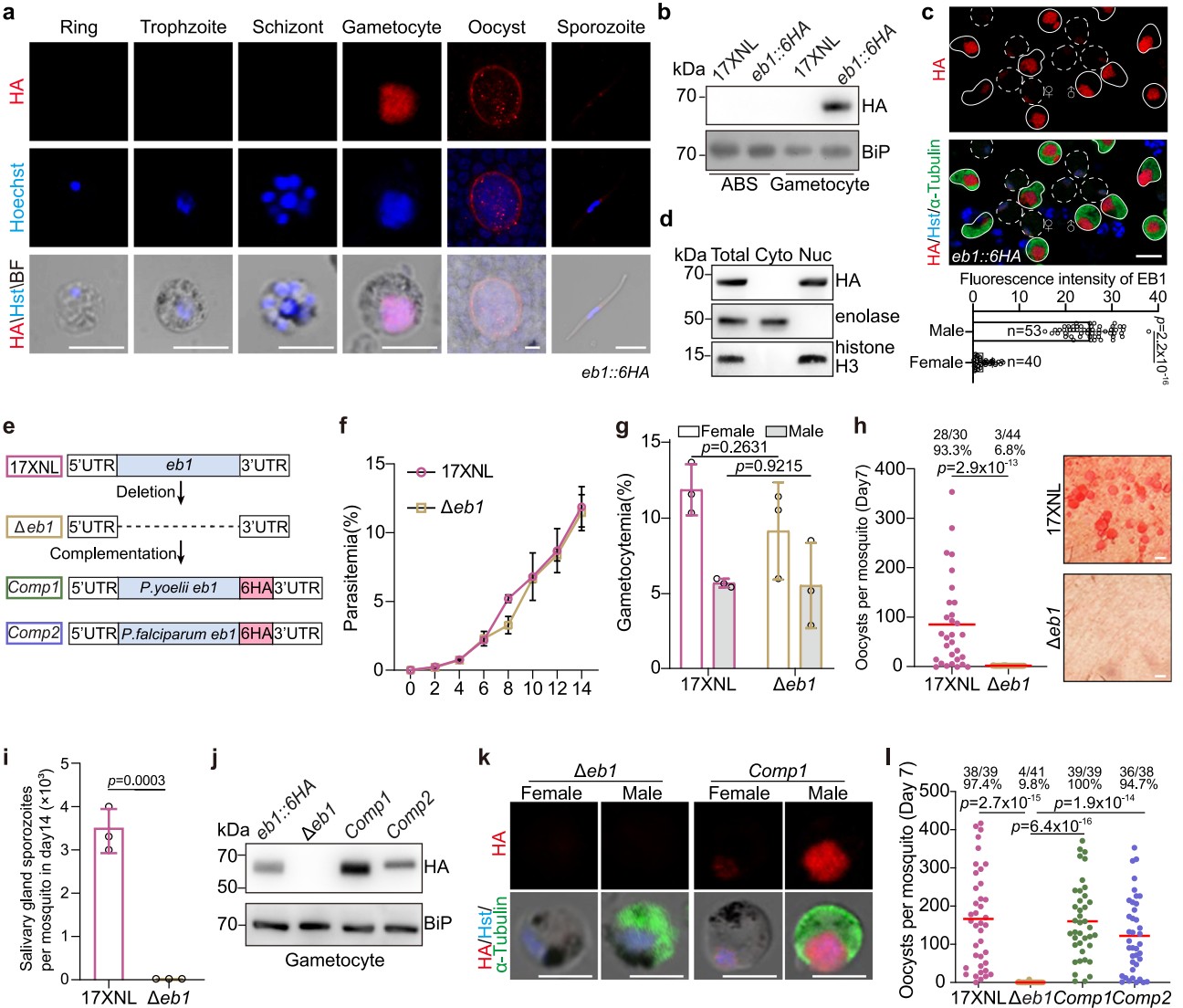

**Fig. 1 | EB1 is expressed in gametocytes and required for parasite transmission in mosquito. a** Immunofluorescence assay (IFA) of EB1 expression in multiple developmental stages of the *eb1::6HA* parasite. The parasites were co-stained with anti-HA antibody and DNA dye Hoechst 33342. Representative for two independent experiments. Scale bar = 5 μm. **b** Immunoblot of EB1 in asexual blood stages (ABS) and gametocytes of the *eb1::6HA* parasite. BiP as loading control. Two independent experiments with similar results. **c** Co-staining the *eb1::6HA* gametocytes with anti-HA antibody and antibody against both α-Tubulin I and α-Tubulin II (male gametocyte marker). Hst, Hoechst 33342. Scale bar = 5 μm. Bottom panel shows the quantification of fluorescence intensity for EB1 protein. *n* is the number of gametocytes measured. Mean ± SD from three independent experiments, two-sided Mann–Whitney *U* test. **d** Immunoblot of EB1 from cytosolic and nuclear fraction of *eb1::6HA* gametocytes. Enolase is stained as a cytosol marker while histone H3 is stained as a nuclear marker. Two independent experiments with similar results. **e** Schematic of the *eb1* gene deletion and gene complementation using the CRISPR-Cas9 method. The whole coding sequence of *eb1* gene was removed in the 17XNL strain, generating the Δ*eb1* mutant. The *eb1* gene from *P. yoelii* and *P. falciparum* respectively fused with a 6HA was introduced back to the *eb1* locus of the Δ*eb1* mutant, generating two complemented strains *Comp1* and *Comp2*. **f** Parasite proliferation at asexual blood stages in mice. Mean ± SD from three mice in each group. Representative for two independent experiments. **g** Male and female gametocyte formation in mice. Mean ± SD from three mice in each group, two-tailed *t*-test. Representative for two independent experiments. **h** Midgut oocyst formation in mosquitos at day 7 post blood feeding. Red horizontal lines show mean value of oocyst numbers, two-sided Mann–Whitney *U* test. x/y on the top is the number of mosquitoes containing oocyst/the number of mosquitoes dissected; the percentage number is the mosquito infection prevalence. Representative results from two independent experiments. Right panel shows the parasite-infected midgut after mercurochrome staining. Scale bar = 50 μm. **i** Salivary gland sporozoite formation in mosquitoes at day 14 post blood feeding. At least 20 infected mosquitoes were counted in each group. Mean ± SD from three independent experiments, two-tailed *t*-test. **j** Immunoblot of HA-tagged EB1 in gametocytes of the complemented parasites *Comp1* and *Comp2*. BiP as loading control. Representative for two independent experiments. **k** IFA of HA-tagged EB1 and α-Tubulin in male and female gametocytes of the *Comp1* strain. Hst, Hoechst 33342. Representative for three independent experiments. Scale bar = 5 μm. **l** Midgut oocyst formation of the *Comp1* and *Comp2* parasite in mosquitoes at day 7 post blood feeding. Red horizontal lines show mean value of oocyst numbers, two-sided Mann–Whitney *U* test. x/y on the top is the number of mosquitoes containing oocyst/the number of mosquitoes dissected; the percentage number is the mosquito infection prevalence. Representative results from two independent experiments.

## EB1 regulates chromosome allocation into male gametes

Selective expression of EB1 in gametocytes suggests a role of EB1 in gametogenesis. Therefore, we measured male gamete formation by counting exflagellation centers (ECs) after stimulation with the xanthurenic acid (XA) at 22 °C in vitro. The Δ*eb1* parasite showed a significant decrease in the number of ECs at 10 min post-activation (mpa) compared to the WT parasites, while both complementation strains, *Comp1* and *Comp2*, markedly restored the EC formation

(Fig. 2a). In contrast, EB1 disruption had no appreciable impact on female gamete formation (Fig. 2b). The nuclear localization and functional importance of EB1 in the male gametogenesis suggest that EB1 is likely involved in genome replication or chromosome segregation. To quantitatively analyze genome replication, we disrupted *eb1* in a *P. yoelii* reporter strain *DFsc7* expressing GFP and mCherry in the male and female gametocytes, respectively[34]. Flow cytometry analyses of the male gametocytes at 8 mpa showed a comparable increase in the DNA content in both the parental *DFsc7* and mutant *DFsc7;Δeb1* parasites (Fig. 2c), indicating normal genome replication. Next, we examined the allocation of duplicated chromosomes in the activated gametocytes. Although individual chromosomes are difficult to be recognized under confocal microscopy, DNA staining still yields clues about global chromosome segregation in the nucleus (Fig. 2d). Spatial separation of the haploid-set chromosomes in the nucleus at 8 mpa was observed in $49.0 ± 4.0\%$ of the WT male gametocytes but only in $10.3 ± 2.1\%$ of *Δeb1* (Fig. 2d), indicating defective chromosome segregation in the absence of EB1. Consequently, only $22.7 ± 5.1\%$ of the *Δeb1* male gametes were nucleated when released from gametocytes, which was significantly lower than that of the WT male gametes ($66.7 ± 3.5\%$; Fig. 2e, f). The *DFsc7;Δeb1* parasites exhibited a similar defect compared to its parental *DFsc7* (Fig. 2g, h). Consistently, *Δeb1* displayed a reduced level of ookinete formation post fertilization in vitro (Fig. 2i). Therefore, EB1 depletion does not affect genome replication but impairs chromosome segregation to daughter male gametes (Fig. 2j).

We also analyzed the assembly of cytoplasmic axonemes, another essential component for male gamete formation. We examined the proportion of male gametocytes developing basal body at 1 mpa, the axonemes coiling around the nucleus at 8 mpa, and the flagellums budding outward at 15 mpa, respectively, all of which were comparable between the WT and *Δeb1* parasites (Supplementary Fig. 2a, b). The characteristic "9 + 2" MT structure detected by transmission electron microscopy further confirmed intact axonemes in the *Δeb1* parasites (Supplementary Fig. 2c).

## EB1 decorates spindle full-length MTs during endomitosis of male gametogenesis

We used IFA to determine the precise localization of EB1 during male gametogenesis. Before activation, EB1 was diffusely distributed in the nucleus. At 1 mpa, EB1 formed a nucleus-associated focus, which was elongated across the nucleus at 2 mpa and further splitted into two foci at the opposite at 3 mpa. After another two rounds of elongation and splitting, eight discrete EB1 foci were formed and associated with the octoploid nucleus or the haploid-separated nucleus (Fig. 3a). The EB1 signals were almost undectable in the exflagellated gametocytes at 15 mpa. We generated another modified strain *eb1::gfp*, in which endogenous EB1 was tagged with GFP. Using this strain, we observed a similar dynamics of cellular localization (Fig. 3b). The localization pattern of EB1 coincided with the mitosis dynamics in male gametogenesis[15,35,36], suggesting possible association of EB1 with spindle. Immunostaining of the *eb1::6HA* male gametocytes with HA and Tubulin antibodies further confirmed spindle-like localization of EB1 (Fig. 3c). Before activation, EB1 was dispersed in the nucleoplasm while Tubulin was mainly distributed in the cytoplasm. After gametocyte activation, EB1 signals overlapped with the spindle MTs throughout three rounds of spindle elongation and splitting (Fig. 3c). The spindle MTs became less recognizable at the later stages because the tubulin fluorescent signal was obscured by the sharply increasing intensity of cytoplasmic axonemes. Throughout the male gametogenesis, EB1 did not associate with the axonemes (Fig. 3c), further supporting its nuclear localization.

We performed ultrastructure expansion microscopy (U-ExM) to analyze the preciselocalization of EB1 at the spindles[37]. U-ExM also revealed co-localization of EB1 with spindles throughout the male gametogenesis (Fig. 3d). The hemispindles that radiated from the spindle pole showed an umbrella-like array of MT fibers and EB1 decorated the full-length MTs from the pole end to the plus end (Fig. 3d). The spindle-specific localization and full-length MT decoration of EB1 were observed in another strain, *eb1::4Myc* (Fig. 3e), in which the endogenous EB1 was tagged with a 4Myc epitope. A schematic diagram for EB1 decoration of spindle MTs in the activated male gametocytes was shown in the Fig. 3f. Additionally, we stained the *eb1::4Myc* gametocytes with antibodies against Myc and polyglutamylated tubulin (PolyE), a marker for stabilized MT[37]. PolyE antibody specifically labeled the axonemes, but not hemispindles (Supplementary Fig. 3a). This phenomenon is consistent with a recent report showing no staining of PolyE on the hemispindles of *P. falciparum* schizonts[37]. In addition, these results suggest a requirement of dynamic MTs for the spindle function during male gametogenesis.

Using EB1 as a spindle marker, we investigated the relative position of the spindles to the adjacent structures including the nuclear envelope, nuclear pore, and basal body. The activated *eb1::6HA* male gametocytes were analyzed via U-ExM. The hemispindle MT fibers were radiated from an NHS dense-stained area which represented the bipartite MTOC (spindle pole and basal body) at the nuclear membrane (Supplementary Fig. 4a). In addition, two double-tagged strains *eb1::6HA;sas4::4Myc* and *eb1::6HA;nup313::4Myc* were generated by tagging endogenous SAS4 (basal body marker[38]) and Nup313 (nuclear pore marker[39]) with 4Myc in the *eb1::6HA* parasites. U-ExM revealed that the spindles were adjacent to the basal bodies in the activated male gametocytes (Supplementary Fig. 4b). In addition, EB1 partially overlapped with the Nup313-labeled nuclear pores, with EB1 locating at the inner side of nuclear envelope (Supplementary Fig. 4c). These results agree well with the current understanding of spindle localization in the male gametogenesis of *Plasmodium*[9]. Thus EB1 could be used to distinguish the spindle MTs from the axoneme MTs during male gametogenesis, especially at the later stages of this process.

## Plasmodium EB1 shows MT-lattice binding affinity

The *Plasmodium* EB1 binds the full-length spindle MTs, suggesting MT-lattice binding affinity, which is distinct from the MT plus-end binding of the canonical EB1[40]. This phenomenon could be caused by the spindle MT and/or EB1 of *Plasmodium*. To figure it out, we chose *P. yoelii* EB1 (PyEB1, 431aa) and *H. sapiens* EB1 (HsEB1, 268aa) for comparison (Fig. 4a and Supplementary Fig. 5). GFP-tagged full-length HsEB1 (HsEB1-FL) and PyEB1 (PyEB1-FL) were transfected in the human MRC5 cell line for heterologous MT co-localization analyses. GFP-tagged HsEB1-FL was co-localized with MT fibers with the expected plus-end accumulation in the transfected cells (Figs. 4b-1), in agreement with previous results[40–42]. In contrast, PyEB1-FL evenly bound the full-length MT (Fig. 4b-4). Staining of the endogenous HsEB1 and transfected PyEB1-FL in the MRC5 cells also revealed the MT plus-end accumulation of HsEB1 and the full-length MT co-localization of PyEB1 (Fig. 4c). These results in the human cells suggest that MT-lattice binding is an intrinsic property of PyEB1, but not due to the parasite MT.

To further validate the MT-lattice binding of PyEB1, we purified GFP-tagged HsEB1-FL and PyEB1-FL from HEK293T cells and incubated the recombinant protein with the MTs in vitro. Total internal reflection fluorescence (TIRF) microscopy showed that GFP-tagged PyEB1-FL autonomously bound both MT seeds and lattices in vitro, while HsEB1-FL tracked the MT growing-ends (Fig. 4d, e). When co-inoculated with the MTs in vitro, the mCherry-tagged HsEB1 only tracked the MT growing ends while the GFP-tagged PyEB1 bound the full-length MTs (Fig. 4f). We noticed the altered dynamics of MT growth, including decreased MT catastrophe and increased MT rescue, in the presence of PyEB1, suggesting that PyEB1 harbors an MT-stabilizing activity. We performed a titration in PyEB1 concentrations of 0, 15, 30, and 60 nM in the in vitro MT-binding assays and detected MT-stabilizing activity at the concentration of 60 nM (Fig. 4g, h). These results confirm that the

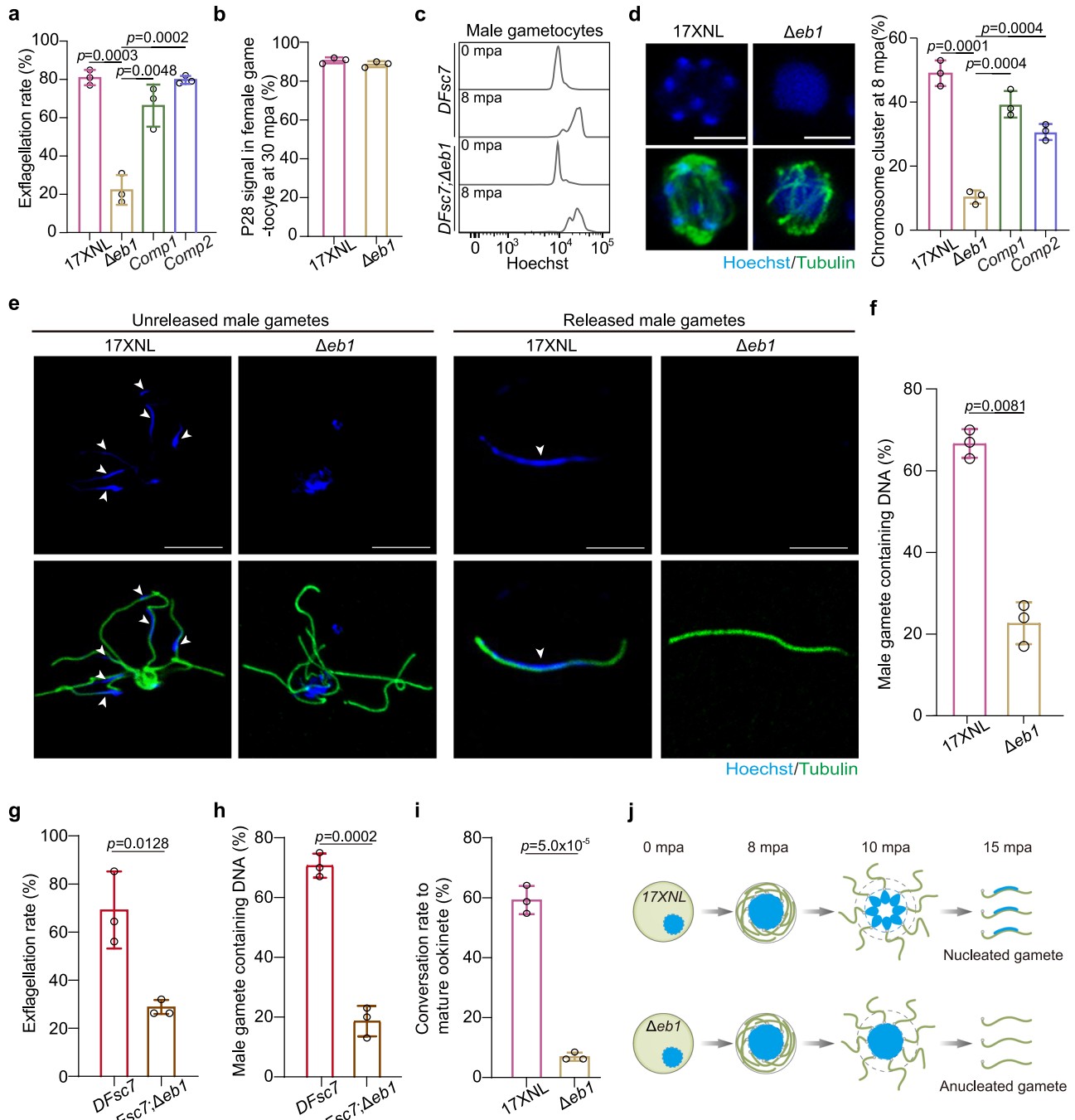

**Fig. 2 | EB1 regulates chromosome allocation into male gametes. a** in vitro exflagellation assay of male gametocytes. Mean ± SD from the infected blood of three mice in each group, two-tailed *t*-test. Representative for three independent experiments. **b** Female gamete formation at 30 min post activation (mpa). The ratio of the number of P28-positive female gametes and zygotes to the number of female gametocytes was calculated. Mean ± SD from three independent experiments, two-tailed *t*-test. **c** Flow cytometry detection of genome DNA content in activated male gametocytes of the *DFsc7* and the *DFsc7;Δeb1* parasites. *DFsc7* is a *P. yoelii* reporter strain expressing GFP and mCherry in male and female gametocytes respectively, and the *DFsc7;Δeb1* is a *DFsc7*-derived EB1-null strain. NAG, non-activated gametocytes. **d** Chromosome segregation in male gametocytes at 8 mpa. The parasites were co-stained with Hoechst 33342 and anti-α-Tubulin antibody. Scale bar = 5 μm. Right panel indicates the percentage of male gametocytes showing chromosome segregation. About 20 male gametocytes were analyzed in each group for each experiment. Mean ± SD from three independent experiments, two-tailed t-test.

**e** Detection of the flagellating and flagellated male gametes containing chromosomes (white arrows). Activated male gametocytes at 15 mpa were co-stained with Hoechst 33342 and anti-α-Tubulin (axoneme microtubule component) antibody. Representative for three independent experiments. Scale bar = 5 μm.
**f** Quantification of male gametes containing chromosomes in **e**. About 50 male gametes were analyzed in each group for each experiment. Mean ± SD from three independent experiments, two-tailed t-test. **g** in vitro exflagellation assay of male gametocytes for *DFsc7* and *DFsc7;Δeb1* parasites. Mean ± SD from three independent experiments, two-tailed *t*-test. **h** Quantification of male gametes containing chromosomes in *DFsc7* and *DFsc7;Δeb1* parasites. Mean ± SD from three independent experiments, two-tailed t-test. **i** In vitro ookinete formation after gamete fertilization. Mean ± SD from three independent experiments, two-tailed *t*-test. **j** Schematic of the defective chromosome segregation caused by EB1 deficiency during male gametogenesis, resulting anucleated male gametes.

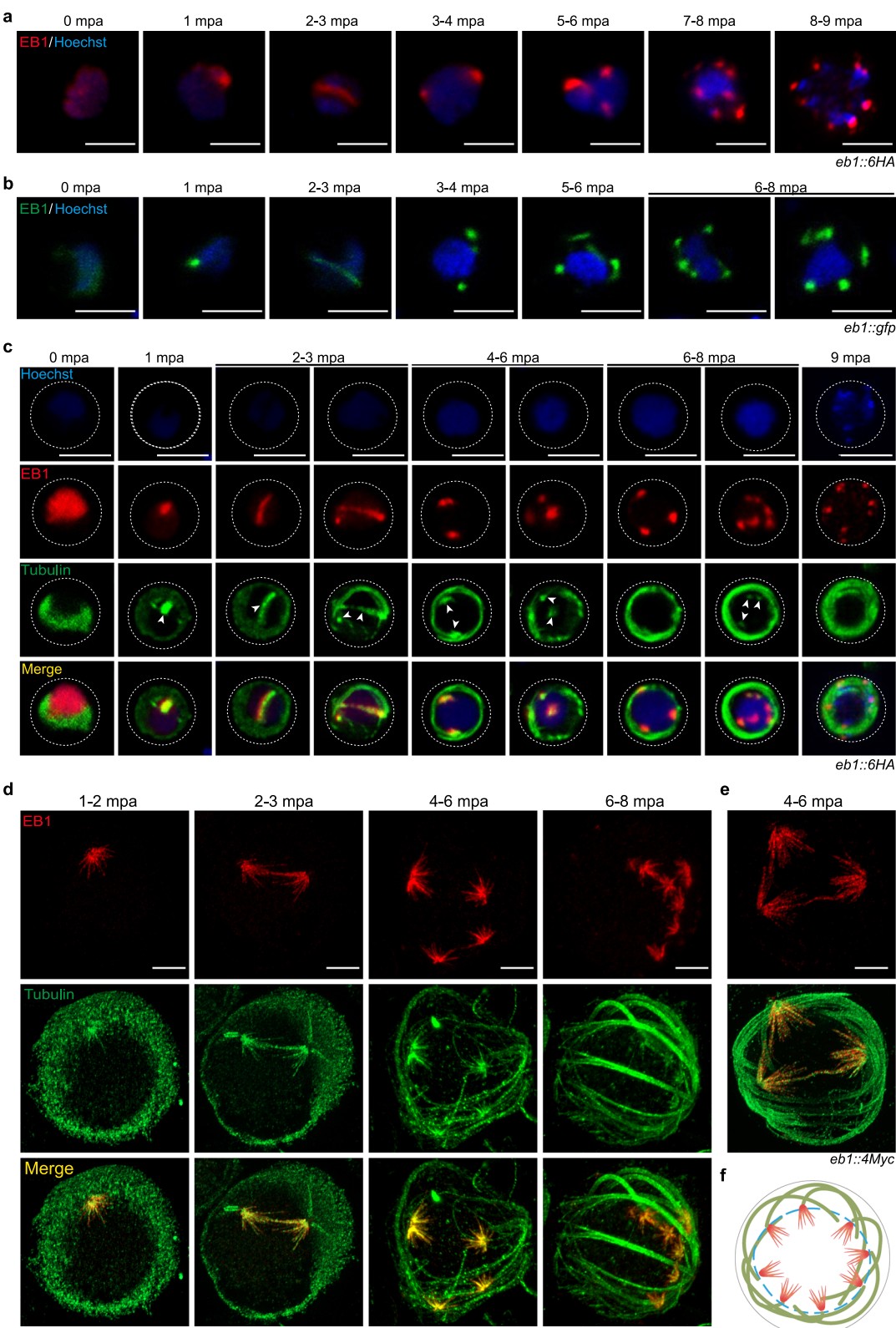

spindle full-length MT binding of PyEB1 in the parasites can be repro-duced in vitro. Thus, PyEB1 is an MT-lattice binding protein and has MT-stabilizing activity.

## CH and linker domains contribute to MT-binding and spindle localization of EB1

We sought to identify PyEB1 sub-region(s) involved in MT-lattice binding. Three partly overlapping fragments (PyEB1-N: 1-177 aa, PyEB1-

M: 140-307 aa, and PyEB1-C: 307-431 aa) were analyzed (Fig. 4a). PyEB1-N contains a N-terminal tail and CH domain; PyEB1-M contains a linker, while PyEB1-C contains CC domain and a C-terminal tail. PyEB1-N was fused at the C-terminus with a leucine zipper (LZ) domain[43] to mimic protein dimerization[44]. When fused with GFP and expressed in the MRC5 cells, both PyEB1-N and PyEB1-M displayed MT localization (Fig. 4b-5, 4b-6) while PyEB1-C did not (Fig. 4b-7). Similar to PyEB1-FL, PyEB1-N and PyEB1-M evenly bound full-length MTs (Fig. 4b). Of note,

**Fig. 3 | EB1 decorates the full-length spindle MTs in activated male gameto-cytes. a** IFA of HA-tagged EB1 in the *eb1::6HA* male gametocytes during gametogenesis. Nuclei are stained with Hoechst 33342. Representative for three independent experiments. Scale bar = 5 µm. **b** Fluorescence microscopy observation of GFP-tagged EB1 in the *eb1::gfp* male gametocytes during gametogenesis. Representative for three independent experiments. Scale bar = 5 µm. **c** Co-staining the *eb1::6HA* male gametocytes with antibodies against HA and *α*-Tubulin during gametogenesis. Hoechst, Hoechst 33342. White arrows indicate the mitotic spindle. Representative for three independent experiments. Scale bar = 5 µm.
**d** Ultrastructure expansion microscopy (U-ExM) of EB1 and *α*-Tubulin expression in

the *eb1::6HA* gametocytes during gametogenesis. Parasites were co-stained with antibodies against HA and *α*-Tubulin. Representative for three independent experiments. Scale bar = 5 µm. **e** U-ExM of EB1 and *α*-Tubulin expression in the *eb1::4Myc* gametocytes at 4-6 mpa. The parasites were co-stained with antibodies against Myc and *α*-Tubulin. Representative for three independent experiments. Scale bar = 5 µm. **f** Model showing EB1 (red) binding the full-length MTs of eight hemispindles in activated male gametocytes. The hemispindles were indicated in red color, the axonemes in green color, and the nuclear membrane was indicated as the blue dashed line.

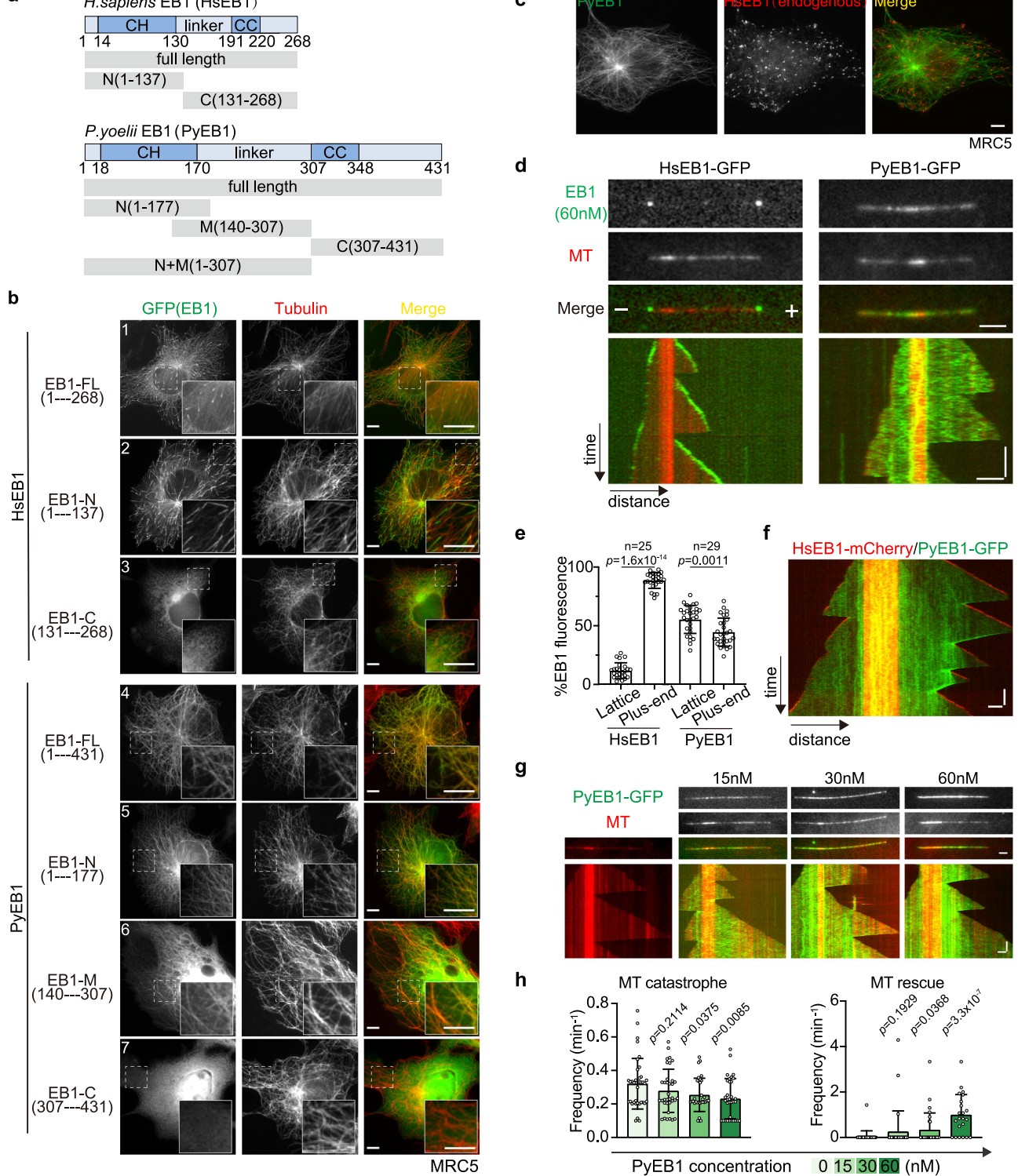

**Fig. 4 | *Plasmodium* EB1 shows MT lattice affinity. a** Protein domain organization of *H. sapiens* EB1 (HsEB1) and *P. yoelii* (PyEB1). Full-length proteins and fragments of HsEB1 and PyEB1 were tested for MT-localization and MT-binding activity. CH: calponin homology domain, CC: coil-coiled domain, linker: the region between CH and CC. The numbers indicate the position of amino acids. **b** IFA of GFP-tagged EB1 proteins (green) and MTs (red) in the human cell line MRC5. Full-length proteins and N-terminus fragments of HsEB1 and PyEB1 were fused with GFP at the C-terminus, M or C-ternius fragments of HsEB1 and PyEB1 were fused with GFP at the N-terminus and transiently expressed in the MRC5. Insets show enlargements of the boxed areas. Representative for three independent experiments. Scale bar = 5 μm. **c** IFA of GFP-tagged PyEB1 (green) and endogenous HsEB1 (red) in the MRC5 cells. Full-length PyEB1 were fused with GFP at the C-terminal and transiently expressed in the MRC5. Representative for three independent experiments. Scale bar = 5 μm. **d** in vitro MT binding of proteins detected by total internal reflection fluorescence (TIRF) microscopy. The GMPCPP-stabilized MT (red) was used as nucleation seed for MT (red) growth in the presence of GFP-tagged HsEB1 or PyEB1 (green). Representative images and kymographs showed MT growth-end tracking of HsEB1 (left panel) and full-length MT binding of PyEB1 (right panel). Note that MT growth in vitro occurs in

both MT-plus and -minus ends. Representative for three independent experiments. Horizontal scale bars: 2 μm; vertical scale bars: 1 min. **e** Quantification of GFP fluorescence signal for HsEB1 and PyEB1 along MT in **d**. Relative intensities of GFP signal on the MT lattice and MT plus-end segments were measured. n is the number of MTs measured in each group. Mean ± SD from three independent experiments, two-sided Mann–Whitney *U* test. **f** in vitro MT binding of HsEB1 and PyEB1 detected by TIRF. The GMPCPP-stabilized MT (red) was used as nucleation seed for MT growth in the presence of both mCherry-tagged HsEB1 and GFP-tagged PyEB1 (60 nM used). Note that MT growth in vitro occurs in both MT-plus and -minus ends. Representative images and kymographs showed MT growth-end tracking of HsEB1 and full-length MT binding of PyEB1. Representative for three independent experiments. Horizontal scale bars: 2 μm; vertical scale bars: 1 min. **g** Representative images and kymographs showing MT-stabilizing activity of PyEB1. Horizontal scale bars: 2 μm; vertical scale bars: 1 min. **h** Quantification of MT dynamic events in **g**, including MT shrinkage, catastrophe, and rescue in the presence of a titration of PyEB1. Mean ± SD from three independent experiments, two-tailed *t*-test.

the HsEB1-N fragment (1-137 aa, containing the N-terminal tail and CH domain) retained the MT plus-end tracking property as HsEB1-FL did (Fig. 4b-2), while HsEB1-C (131-268 aa, containing the linker, CC domain, and C-terminal tail) distributed diffusely in the cytosol (Fig. 4b-3). We attempted to analyze the individual PyEB1-N and PyEB1-M for the in vitro MT-binding assay but failed to purify the recombinant proteins. Instead, we obtained recombinant GFP-tagged PyEB1-N + M (1-307 aa) (Fig. 4a and Supplementary Fig. 6a). This fragment showed co-localization with the full-length MT in the MRC5 cells (Supplementary Fig. 6b) and bound both MT seeds and lattices in vitro (Supplementary Fig. 6c), behaving like PyEB1-FL. These results indicated that the CH and linker of PyEB1 both possesses the MT-binding activity. In addition, they function together to confer the MT-lattice binding affinity of PyEB1, distinct from the canonical EB1 proteins which bind the MT plus-end via CH domain.

We next investigated the roles of MT-binding domains in spindle localization of PyEB1 and in chromosome segregation during male gametogenesis. The genomic sequence encoding EB1 N-terminal tail, CH, and linker domain, were deleted from the parental parasite *eb1::6HA*, respectively (Fig. 5a). Domain truncation had no remarkable effect on the protein levels of PyEB1 (Fig. 5b). Truncation of either CH or linker dramatically impaired spindle localization of PyEB1 (parasites with EB1 spindle localization: 80.0 ± 7.6% in *eb1::6HA*; 5.3 ± 3.1% in *ΔTrunc2*; 34.7 ± 19.1% in *ΔTrunc3*) (Fig. 5c). CH- or linker-truncated EB1 was distributed in the nucleoplasm (Fig. 5c), which is further confirmed by U-ExM analyses (Fig. 5d). Furthermore, truncation of CH and linker completely ablated PyEB1 spindle localization (Fig. 5c). Therefore, both CH and linker, two independent MT-binding domains, are critical for spindle localization of PyEB1. Consistently, loss of single CH or linker impaired chromosome segregation in the gametocytes at 8 mpa while loss of both domains additively abrogated chromosome segregation (Fig. 5e). Interestingly, truncation of N-terminal tail seemed to have little effect on the spindle localization of PyEB1 (Fig. 5c), but resulted in decreased chromosome segregation (Fig. 5e), suggesting that the N-terminal tail is not required for MT binding but functions in chromosome segregation.

## EB1 is critical for spindle structure

Since EB1 decorates the full-length MTs of hemispindles during the three rounds of mitosis, we reasoned that EB1 may regulate spindle assembly, structure, and/or function. We examined the spindle assembly during early male gametogenesis. In both the WT and *Δeb1* male gametocytes, tubulin staining at 1 mpa revealed a characteristic nucleus-associated spindle dot, which further elongated across nucleus at 2 mpa (Supplementary Fig. 7). Quantification indicated no marked effect on the early spindle assembly after EB1 disruption

(Supplementary Fig. 7). We sought to analyze spindle structure during later gametogenesis using similar staining methods but failed due to the undistinguishable MT fluorescence signals representing both spindles and axonemes. Alternatively, male gametocytes at 8 mpa were co-stained with antibodies against Tubulin and PolyE, the latter of which specifically recognizes the axonemal MTs (Supplementary Fig. 3b). This immunostaining combined with U-ExM enabled us to visualize spindles at high resolution in the activated male gametocytes at later stages, where the spindles were Tubulin-positive and PolyE-negative (Supplementary Fig. 3b). WT parasites showed an umbrella-like array of hemispindle MTs (Fig. 6a). In contrast, the hemispindles in *Δeb1* appeared to be smaller and shorter. We quantified the hemispindle MTs number and MT length. Approximately 7.8 ± 2.1 (*n* = 21) MTs per hemispindle were detected in the WT gametocytes at 8 mpa, which was significantly decreased in the *Δeb1* gametocytes (4.4 ± 1.8 MTs per hemispindle, *n* = 26) (Fig. 6b). The MT length was 1.0 ± 0.4 μm (*n* = 164) in the WT parasites and 0.5 ± 0.3 μm (*n* = 114) in *Δeb1* (Fig. 6c). In addition, we measured the maximum spindle spread angle[45]. The activated EB1-null gametocytes exhibited a smaller spindle spread angle than the WT gametocytes (Fig. 6d). Therefore, EB1 is dispensable for early spindle assembly but is important for the structure of later stage hemispindles.

## Spindle laterally connects kinetochores with a requirement of EB1

The primary role of spindle is to connect the kinetochores for directing the chromosome segregation in eukaryotic cells. To date, the spatio-temporal relationship between spindle and kinetochore in *Plasmodium* gametogenesis remains largely elusive. Since the EB1 deficiency causes abnormal spindles, we speculated a defective spindle-kinetochore attachment in the EB1-null male gametocytes. To test it, we tagged the outer kinetochore protein Ndc80 (PY17X_1116900)[15] with 4Myc in the *eb1::6HA* parasite and obtained a double-tagged parasite clone, *eb1::6HA;ndc80::4Myc* (*DTS1*) (Supplementary Fig. 8a). IFA showed that Ndc80 was expressed in all the proliferative stages of parasites (Supplementary Fig. 8a), consistent with previous results[15]. In the gametocytes, both EB1 and Ndc80 were diffuse in the nucleoplasm (Supplementary Fig. 8a). After activation, Ndc80 was concentrated into foci and dynamically co-localized with EB1 throughout the gametogenesis (Supplementary Fig. 8a). Co-immunoprecipitation detected EB1 interaction with Ndc80 in the activated gametocytes, but not in the non-activated gametocytes (Supplementary Fig. 8b, c), further confirming the spindle-kinetochore attachment in gametocytes after activation. U-ExM analyses of the activated *DTS1* male gametocytes distinguished Ndc80 positioning relative to EB1 (Fig. 7a). Nearly all the individual Ndc80 foci were associated with spindles

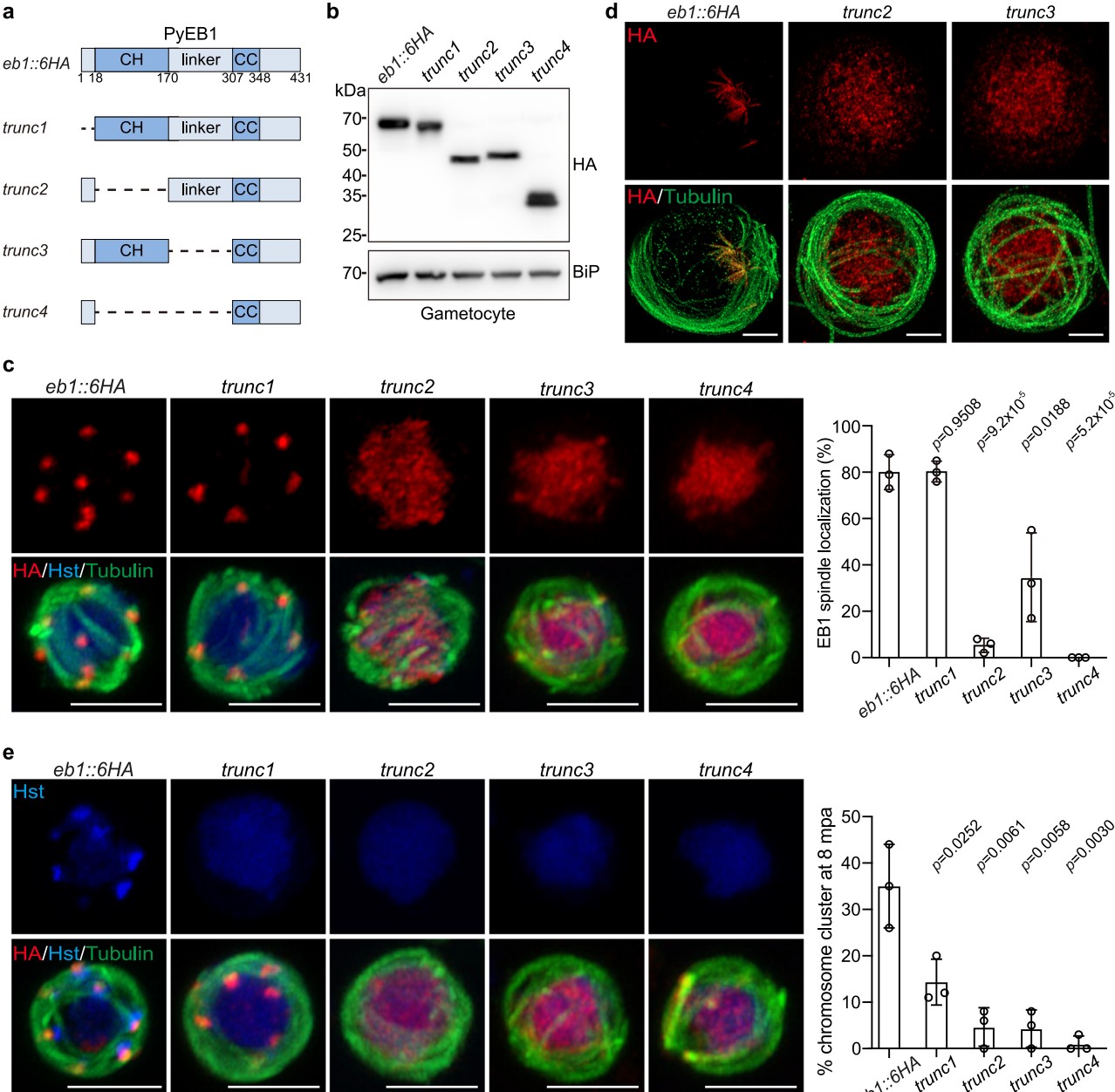

**Fig. 5 | CH and linker domains contribute to MT-lattice binding and spindle localization of EB1. a** Diagram of domain truncation in endogenous PyEB1 from the parental parasite *eb1::6HA*, generating four truncated strains *trunc1*, *trunc2*, *trunc3*, and *trunc4*. **b** Immunoblot of HA-tagged EB1 in the gametocytes of the *eb1::6HA*, *trunc1*, *trunc2*, *trunc3*, and *trunc4* parasites. BiP as loading control. Representative for two independent experiments. **c** IFA of HA-tagged EB1 (red) and α-Tubulin in male gametocytes at 8 mpa. Hst, Hoechst 33342. Scale bar = 5 μm. Right panel showed the percentage of parasites displaying spindle localization of EB1. About 20 parasites were analyzed in each group for each experiment. Mean ± SD from three independent experiments, two-tailed *t*-test. **d** U-ExM of HA-tagged EB1 and α-Tubulin expression in male gametocytes at 8 mpa. Scale bar = 5 μm. Representative for two independent experiments. **e** Chromosome segregation in male gametocytes at 8 mpa. The parasites were stained with Hoechst 33342, anti-HA and anti-α-Tubulin antibodies. Scale bar = 5 μm. Right panel indicates the percentage of male gametocytes showing chromosome segregation. Mean ± SD from three independent experiments, two-tailed *t*-test.

throughout the gametogenesis (Fig. 7a). Strikingly, the Ndc80 foci displayed an unusual lateral localization to the spindle MTs (Fig. 7a, b), distinct from the end-on attachment of kinetochores to spindle MTs in the bipolar mitosis[46,47]. To further confirm the lateral positioning of kinetochores to spindle MTs, three other kinetochore proteins including SPC24 (PY17X_1444800), SPC25 (PY17X_1364500), and AKiT1(PY17X_0624000) were included in our analyses. SPC24 and SPC25, similar to Ndc80, are the subunits of the outer kinetochore complex NDC80[15]. AKiT1 is a recently identified *Plasmodium* inner kinetochore protein[16]. Three double-tagged strains *eb1::6HA*; spc24::4Myc (*DTS2*), *eb1::6HA*; spc25::4Myc (*DTS3*), and *eb1::6HA*;

*Akit1::4Myc* (*DTS4*), were generated from the *eb1::6HA* parasites, respectively (Supplementary Fig. 8d–f). Similarly as Ndc80 did, the SPC24, SPC25, and AKiT1 displayed lateral localization to the spindle MTs throughout the endomitosis, shown by both IFA (Supplementary Fig. 8d–f) and U-ExM (Supplementary Fig. 9a–c). These results demonstrate that kinetochores establish connection with spindles from the beginning of endomitosis and maintain the connection throughout. In addition, the EB1-decorated spindle MTs are laterally associated with the kinetochores.

We next asked whether EB1 is required for the spindle-kinetochore attachment. We removed *eb1* gene in the *DTS1* strain

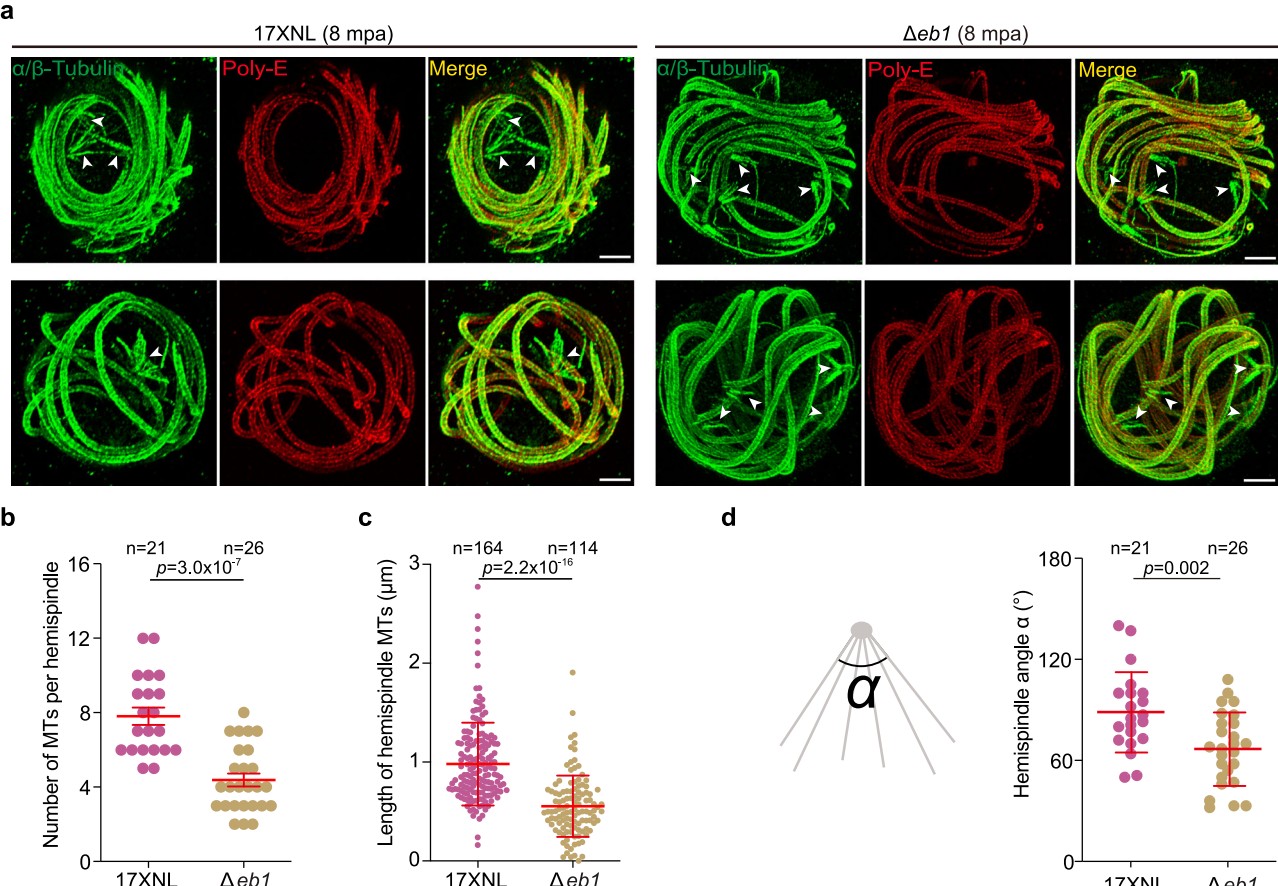

**Fig. 6 | EB1 is critical for spindle structure. a** U-ExM of hemispindle in 17XNL and Δ*eb1* gametocytes at 8 mpa. The parasites were stained with the antibodies against α/β-Tubulins and the glutamylated tubulin (PolyE). PolyE antibody specifically recognized the axonemal MTs while the spindle MTs (white arrows) were Tubulin-positive and PolyE-negative, which enables visualizing spindles. Full section projections of fluorescent signal in parasites were collected. Representative maximum intensity projection (MIP) images were shown. Scale bar = 5 μm. Three independent experiments with similar results. **b** Quantification of MT number per hemispindle in **a**. Mean ± SD from three independent experiments, two-tailed *t*-test. n is the number of hemispindles measured. **c** Quantification of hemispindle MT length in **a**. Mean ± SD from three independent experiments, two-tailed *t*-test. *n* is the number of hemispindles MT. **d** Quantification of the hemispindle angle in **a**. Left panel is a schematic of the hemispindle angle measured. Mean ± SD from three independent experiments, two-tailed *t*-test. *n* is the number of hemispindles measured.

and obtained the EB1-null mutant, *DTS1;Δeb1*. EB1 disruption seemed to have no notable effect on Ndc80 protein level in the male gametocytes. However, the activated *DTS1;Δeb1* male gametocytes exhibited significantly reduced spindle-clustering of kinetochore Ndc80 foci compared to the parental *DTS1* during the three rounds of endomitosis (Fig. 7c). Consistently, U-ExM revealed that the Ndc80 dots lost spindle-clustering and were dispersed throughout the nucleoplasm in the activated *DTS1;Δeb1* male gametocytes (Fig. 7d). To further confirm the defective spindle-kinetochore attachment in the absence of EB1, we removed *eb1* gene in the strains *DTS2* and *DTS3*, respectively and observed similar defects in the spindle-kinetochore attachment during the male gametogenesis (Supplementary Fig. 10a, b).

**Spindle localization of EB1 is independent of kinetochore**
Throughout the multiple rounds of endomitosis, the spindles dynamically and closely associate with the kinetochores (Fig. 7a and Supplementary Figs. 8 and 9). We investigated whether the kinetochores in turn influence the spindle formation and/or the spindle localization of EB1. To date, no functional elucidation of NDC80 has been reported in the male gametogenesis of *Plasmodium*. We attempted to disrupt *ndc80* in the *P. yoelii* parasites but failed, suggesting an essential role of Ndc80 in the asexual blood stage development. This functional essentiality of Ndc80 is consistent with its expression at the asexual blood stage of *P. yoelii* (Supplementary Fig. 8a) and *P. berghei*[15], as well as a recent function study of Ndc80 in

the *P. falciparum* schizogony[48]. We used a promoter swap method to replace 801 bp of endogenous *ndc80* promoter sequence in the double-tagged strain *DTS1* with that (1826 bp) of *clag1* gene (PY17X_1402200) (Fig. 8a), whose transcripts are expressed at the asexual stages, but absent in the gametocytes and mosquito stages[49]. Correct modification in the resulting mutant parasite *ndc80kd* was confirmed by PCR. The promoter replacement allowed Ndc80 expression in asexual blood stage at a level comparable with that of the parental strain *DTS1*, but significantly reduced Ndc80 expression in the gametocytes either before or after activation, which was confirmed by immunoblot (Fig. 8b) and IFA (Fig. 8c). Ndc80 depletion did not appear to affect the spindle formation after gametocyte activation (see the HA-tagged EB1 signal in Fig. 8c, d). EB1 expression and localization showed no difference in the activated gametocytes between *DTS1* and *ndc80kd* (Fig. 8c, d), indicating that EB1 does not rely on Ndc80 for spindle localization. However, the *ndc80kd* parasites significantly decreased the activated male gametocytes showing typical chromosome segregation at 8 mpa (Fig. 8e), consistent with the notion that NDC80 functions in the spindle-kinetochore attachment. The *ndc80kd* parasites produced fewer nucleated male gametes in vitro compared to the parental strain *DTS1* (Fig. 8f). In mosquitoes fed with *ndc80kd* parasite-infected mouse blood, no oocysts were detected in the mosquito midguts (Fig. 8g). Therefore, the spindle localization of EB1 is independent of Ndc80. However, Ndc80 depletion in the gametocytes leads to chromosome

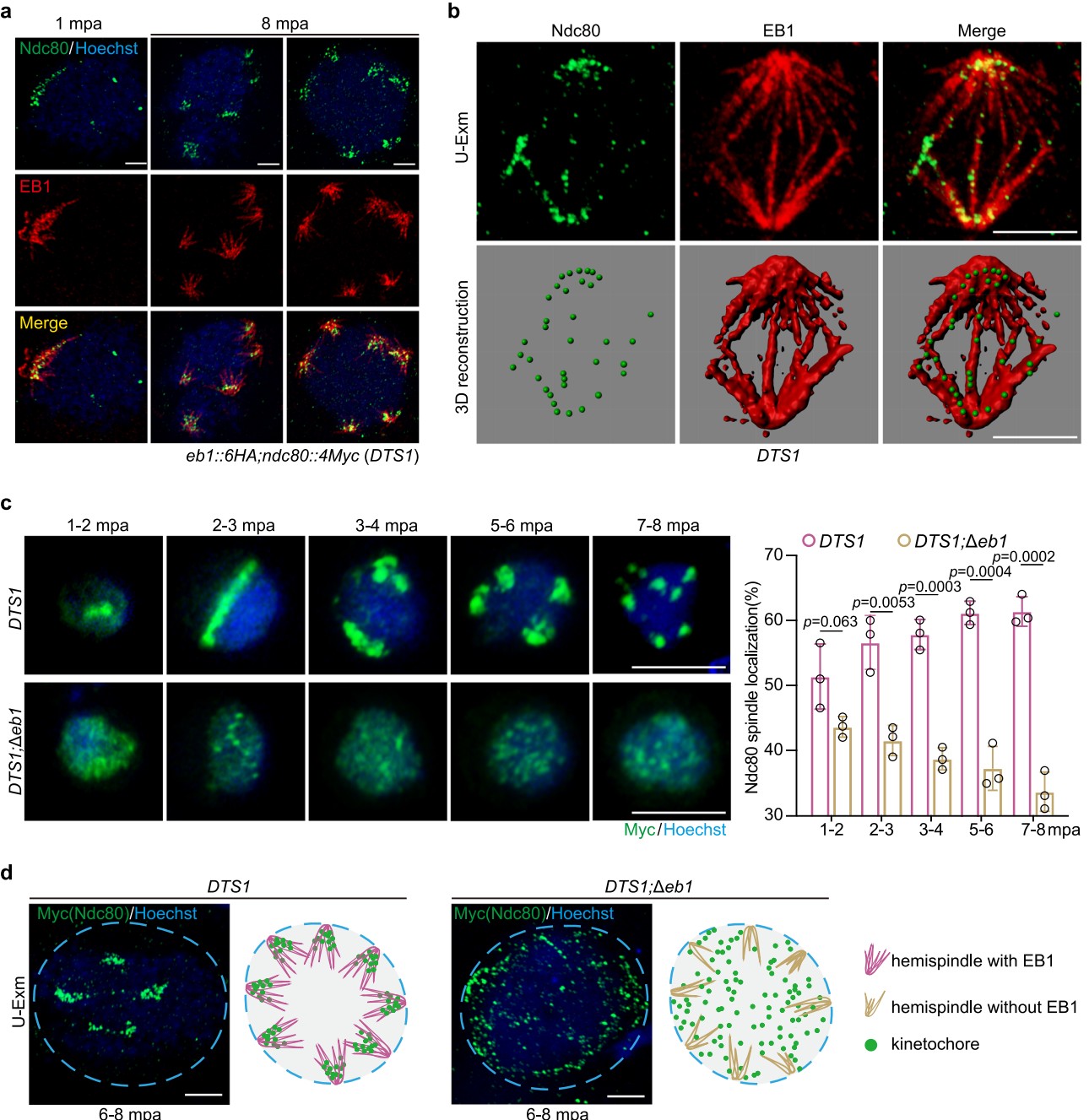

**Fig. 7 | Kinetochores connect spindle laterally and this connection requires EB1. a** U-ExM of EB1 and kinetochore protein Ndc80 in gametocytes of the double-tagged strain *eb1::6HA;ndc80::4Myc* (*DTS1*) along gametogenesis. Parasites were co-stained with Hoechst 33342, anti-HA and anti-Myc antibodies. Scale bar, 5 μm. Representative images from three independent experiments with similar results. **b** Representative full section projections of fluorescent signal in gametocytes from **a**. The spindle was labeled by EB1 and kinetochore was labeled by Ndc80. The images in the lower panels show the 3D surface topology reconstruction of fluorescence signals. **c** IFA of Ndc80 in male gametocytes of the *DTS1* and *DTS1;Δeb1* parasites along gametogenesis. *DTS1;Δeb1* is a *DTS1*-derived EB1-null strain. Parasites were co-stained with anti-Myc antibody and Hoechst 33342. Scale bar = 5 μm. Right panel is the quantification of male gametocytes showing the spindle localization of Ndc80. Mean ± SD from three independent experiments, two-tailed *t*-test. **d** U-ExM of Ndc80 in male gametocytes of the *DTS1* and *DTS1;Δeb1* parasites at 6–8 mpa. The parasites were stained with anti-Myc antibody and Hoechst 33342. Representative for three independent experiments. Scale bar = 5 μm. The schematics show the spatial relationship between hemispindle and kinetochores during male gametogenesis with or without EB1.

segregation defect, resembling that caused by EB1 deficiency in the male gametogenesis.

**Serine 15 phosphorylation of EB1 is critical for spindle-kinetochores attachment**
Recent phosphoproteomic studies have revealed phosphor-regulation of the mitosis- and microtubule-related proteins during the male

gametogenesis of *P. berghei* and *P. falciparum*[50,51]. Serine 15 (S15) in the N-terminal tail of EB1 was independently detected to be phosphorylated in the activated gametocytes of both *P. berghei* and *P. falciparum*[50,51]. S15 residue is conserved among different *Plasmodium* species (Fig. 9a and Supplementary Fig. 1). Whether S15 phosphorylation plays a role in EB1 cellular localization or function remains undetermined. To investigate the potential role of S15 phosphorylation, we

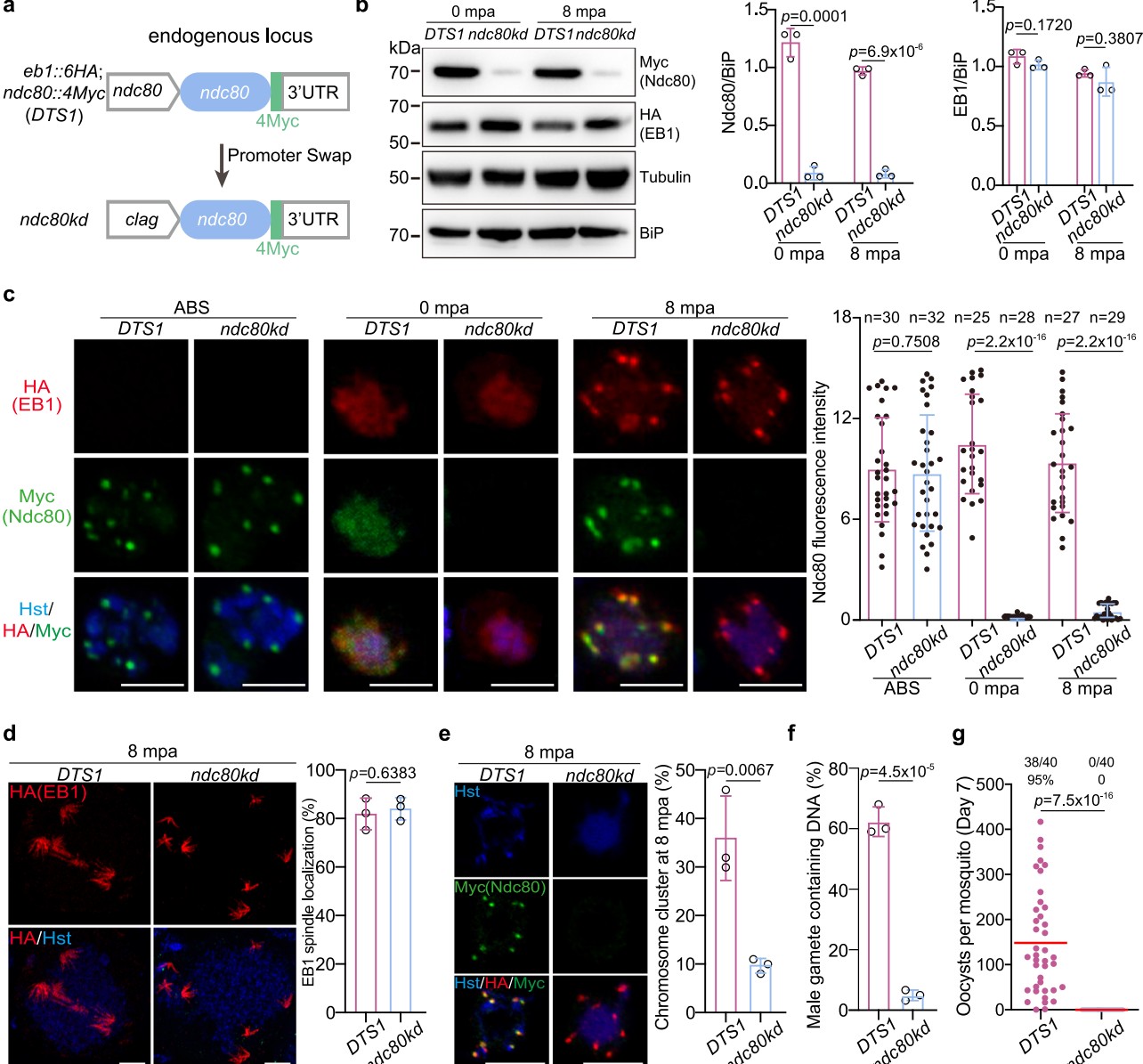

**Fig. 8 | Spindle localization of EB1 is independent of NDC80. a** Schematic of CRISPR/Cas9-mediated promoter swap in the *ndc80* gene locus for depleting the *ndc80* gene expression in gametocytes. The promoter (801 bp) of *ndc80* gene in the *DTS1* parasite was replaced with the promoter (1,826 bp) of the *clag* gene, generating the *ndc80kd* mutant clone. **b** Immunoblot of Myc-tagged Ndc80 and HA-tagged EB1 in gametocytes of the *DTS1* and *ndc80kd* parasites at 0 and 8 mpa. BiP and α-Tubulin as loading control. Right panels show the quantification of Ndc80 and EB1 protein level after normalizing with that of BiP. Mean ± SD from three independent experiments, two-tailed *t*-test. **c** IFA confirming the depleted expression of Ndc80 in the *ndc80kd* male gametocytes. Asexual blood stage (ABS) and gametocytes at 0 and 8 mpa of the *DTS1* and *ndc80kd* parasites were stained with Hoechst 33342 and antibodies against HA and Myc. Scale bar = 5 μm. Right panel shows the quantification of fluorescence intensity for Ndc80 protein. *n* is the number of gametocytes measured. Mean ± SD from three independent experiments, two-tailed *t*-test. **d** U-ExM of EB1 in male gametocytes at 8 mpa. The parasites were stained with Hoechst 33342 and anti-HA antibody. Scale bar = 5 μm. Right panel shows the percentage of parasites with spindle localizing of EB1. Mean ± SD from three independent experiments, two-tailed t-test. **e** Chromosome segregation in male gametocytes at 8 mpa. The parasites were co-stained with Hoechst 33342 and anti-Myc antibody. Scale bar = 5 μm. Right panel indicates the percentage of parasites with chromosome clustering. Mean ± SD from three independent experiments, two-tailed *t*-test. **f** Quantification of the male gamete containing chromosomes in *DTS1* and *ndc80kd* parasites. Mean ± SD from three independent experiments, two-tailed *t*-test. **g** Midgut oocyst formation of the *DTS1* and *ndc80kd* parasite in mosquitoes at day 7 post blood feeding. Red horizontal lines show mean value of oocyst numbers, Mann–Whitney *U* test. x/y on the top is the number of mosquitoes containing oocyst/the number of mosquitoes dissected; the percentage number is the mosquito infection prevalence. Two independent experiments with similar results.

replaced EB1 S15 with Alanine (A) in the *DTS1* parasites, generating the mutant clone *S15A* (Fig. 9b). S15A substitution had no obvious effect on the protein level of EB1 (Fig. 9c). We generated antiserum targeting a synthetic EB1 peptide mimicking phosphorylated S15. This antiserum recognized cell lysates of the activated gametocytes but not the non-activated gametocytes of the *DTS1* strain (Fig. 9c). In contrast, no band

was detected in the activated *S15A* gametocytes using the antiserum (Fig. 9c). These results confirmed EB1 S15 phosphorylation in the activated male gametocytes of *P. yoelii*, similar to those reported in *P. berghei* and *P. faciparum*[50,51]. The *S15A* gametocytes showed spindle localization of EB1 after activation (Fig. 9d, e). Consistent with this, heterologous expression of the GFP-tagged EB1-S15A protein exhibited

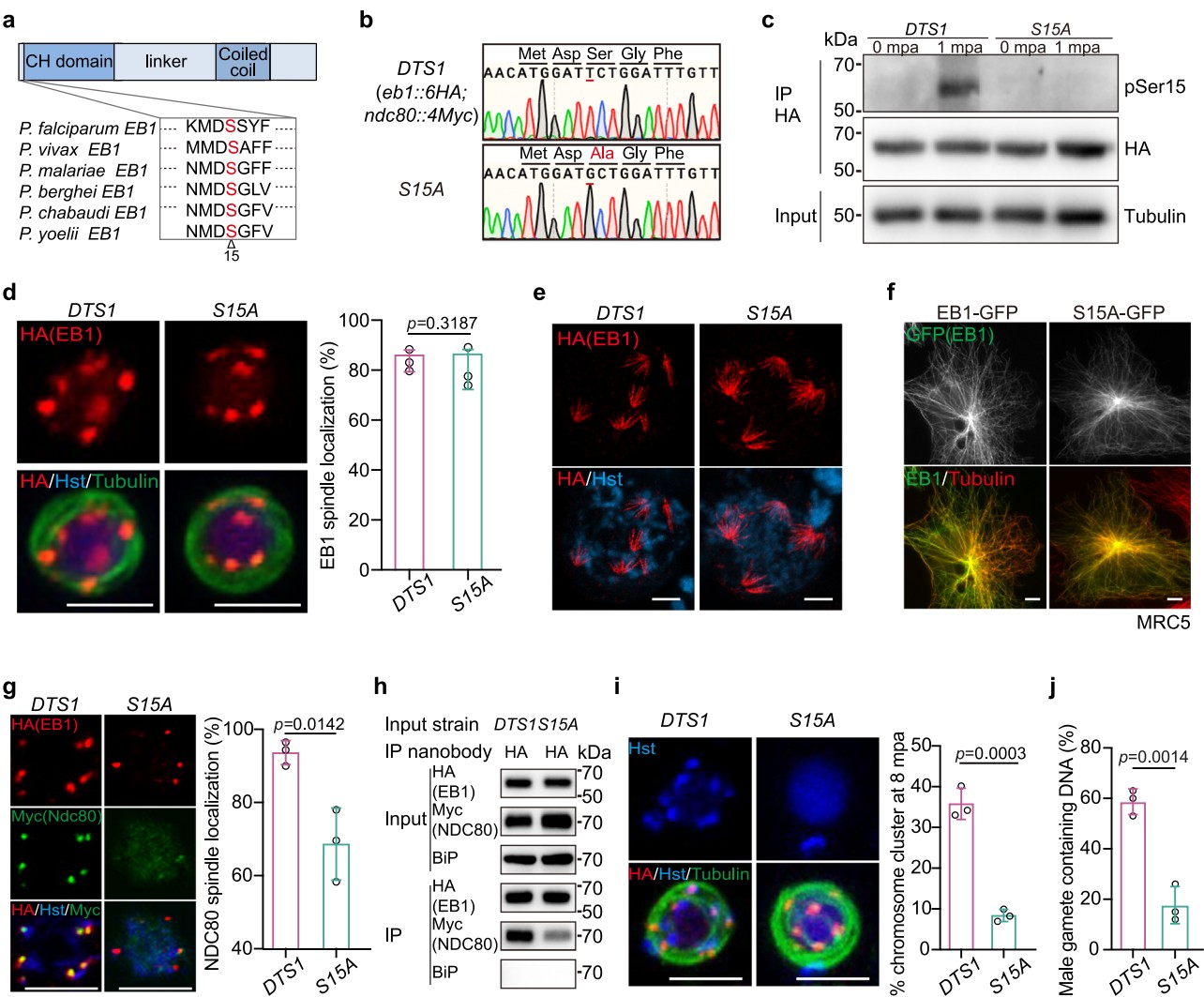

**Fig. 9 | Phosphorylation of EB1 is critical for spindle-kinetochores attachment.**
**a** Potential phosphorylated residue serine 15 (S15, marked with triangle) in the N-terminal tail of EB1, which is conserved among six *Plasmodium* species.
**b** Characterization of mutant parasite with S15 replaced with alanine (A) in endogenous EB1 from the parental parasite *DTS1*. DNA sequencing confirming S15A substitution in the resulting *S15A* mutant. **c** Immunoblot detecting EB1 expression and EB1 S15 phosphorylation (pSer15) in the *DTS1* and *S15A* gametocytes at 0 and 1 mpa. HA-tagged EB1 was immunoprecipitated using anti-HA nanobody and the pSer15 was detected using a home-made antiserum that specifically recognizes EB1 pS15. Tubulin as loading control. Representative for three independent experiments. **d** Co-staining the *DTS1* and *S15A* male gametocytes at 8 mpa with antibodies against HA and α-Tubulin. Hst, Hoechst 33342. Scale bar = 5 μm. Right panel showed the percentage of parasites with spindle localization of EB1. Mean ± SD from three independent experiments, two-tailed *t*-test. **e** U-ExM of EB1 and chromosomes in the *DTS1* and *S15A* gametocytes at 8 mpa. The parasites were co-stained with anti-HA antibody and Hoechst 33342. Representative for three

independent experiments. Scale bar = 5 μm. **f** IFA of GFP-tagged EB1 (green) and MTs (red) in the human MRC5 cells. PyEB1 and S15A mutant proteins were fused with GFP at the C-terminal and transiently expressed in the MRC5. Representative for three independent experiments. Scale bar = 5 μm. **g** Co-staining the *DTS1* and *S15A* male gametocytes at 8 mpa with antibodies against HA and Myc. Right panel showed the percentage of parasites with spindle localization of Ndc80. About 40 parasites were analyzed in each group for each experiment. Mean ± SD from three independent experiments, two-tailed *t*-test. **h** Co-immunoprecipitation of EB1 and Ndc80 in the *DTS1* and *S15A* gametocytes at 8 mpa. Anti-HA nanobody was used for immunoprecipitation. BiP as loading control. Representative for two independent experiments. **i** Chromosome segregation in male gametocytes at 8 mpa. The parasites were co-stained with Hoechst 33342 and antibodies against HA and α-Tubulin. Scale bar = 5 μm. Right panel indicates the percentage of parasites with chromosome clustering. Mean ± SD from three independent experiments, two-tailed *t*-test. **j** Quantification of male gametes containing chromosomes. Mean ± SD from three independent experiments, two-tailed *t*-test.

co-localization with MT fibers in the MRC5 cells (Fig. 9f). Notably, the spindle attachment of Ndc80 was detected in 68.6 ± 9.9% of the activated *S15A* male gametocytes compared to the *DTS1* counterpart (93.6 ± 3.3%) (Fig. 9g). Co-immunoprecipitation also detected dramatically reduced association between Ndc80 and EB1 in the activated gametocytes of *S15A* compared to *DTS1* (Fig. 9h). These results indicated that S15 phosphorylation is important for EB1 function in the spindle-kinetochore attachment. Consistently, the activated *S15A* male gametocytes displayed less chromosome segregation (Fig. 9i) and produced fewer nucleated male gametes in vitro (Fig. 9j).

## Discussion

Activated *Plasmodium* male gametocytes undergo three rounds of endomitosis, yielding a polyploidy nucleus containing 8 sets of diploid chromosomes, which are further simultaneously segregated by 8 hemispindles into 8 daughter gametes. This phenomena implies the presence of a non-canonical spindle-kinetochore capture or attachment for chromosome segregation. However, it has not been fully addressed how the hemispindles are regulated and how the spindle MTs are attached to the kinetochores. In this study, we found that the *Plasmodium* EB1 is a novel spindle MT-binding protein and plays

critical roles in spindle structure and spindle-kinetochore attachment for chromosome segregation during male gametogenesis.

It is well established that the eukaryotic EB1 proteins selectively recognize and accumulate at the growing MT plus-end both in vivo and in vitro[20,21,52,53]. EB1 proteins in different species share similar domain organization, consisting of an N-terminal tail, a CH domain, a flexible linker region, a CC domain, and a C-terminal tail. The CH domain directly interacts with the MTs for autonomous plus-end tracking, the CC domain for protein dimerization, while the C-terminal tail for recruitment of other plus-end tracking proteins[54]. Structural evidence has further revealed that EB1 binding with the MT plus-end is determined by the GDP-Pi-bound Tubulin at the plus-end, whereas Tubulin in the MT lattice is GDP-bound[25,26,55]. In this study, under high resolution fulfilled by the U-ExM, we unexpectedly observed that PyEB1 decorated the full-length MTs of hemispindles in the *Plasmodium* male gametogenesis, suggesting MT-lattice binding affinity of PyEB1. This distinctive MT binding of PyEB1 could be due to the relatively short length or different structural confirmation of spindle MTs in the *Plasmodium*. However, this hypothesis is denied by the observation of MT-lattice binding of PyEB1 in mammalian cells and in vitro MT-binding assay. Our results from both in vitro and in vivo, as well as the heterologous expression assays, provide independent evidence that the MT-lattice binding of PyEB1 is intrinsic and is not dependent on the *Plasmodium* MTs. Interestingly, the MT-binding feature can be modulated by single residue mutation at the Q89 in the CH domain of the fission yeast EB1 protein, Mal3[26,56]. Q89A mutation shifts Mal3 MT binding from the plus-end to the entire lattice[26,56], similar to that of PyEB1 observed in our study. However, the Q89 residue is conserved among EB1 proteins from all the eukaryotic organisms (Supplementary Fig. 5, see the Q106 residue of PyEB1), implying that PyEB1 utilizes a different mechanism to achieve the MT-lattice binding compared with the Q89A-mutant Mal3.

Compared with the canonical EB1, PyEB1 has a longer CH domain and a longer linker, the latter of which is relatively less conserved (Supplementary Fig. 5). We analyze the PyEB1 sub-region(s) and identify both the CH domain and linker region displaying the MT-lattice binding. Consistently, both the CH domain and linker region are critical for PyEB1 function in chromosome segregation in the male gametogenesis. Therefore, except the evolutionarily conserved CH domain, the flexible linker region of PyEB1 independently possesses the MT-binding activity. In addition, they function together to confer the MT-lattice binding of PyEB1, distinct from the canonical EB1 which possesses the MT plus-end binding via CH domain only. The structural basis of MT-lattice recognition by PyEB1 is still unknown. To understand the key structural difference between PyEB1 and canonical EB1, future studies into an atomic resolution structure of PyEB1 on the MT-lattice as well as the relationship of the CH domain and linker region in MT-lattice binding of PyEB1 will be required.

The nucleoplasm localization of PyEB1 in both male and female gametocytes (Fig. 1c) indicates the presence of a potential nuclear localization signal (NLS) in PyEB1. It has been reported that the *Toxoplasma gondii* EB1 (TgEB1) resides in the nucleus mediated by a C-terminal NLS[57]. In addition, an EB1 variant from the *Arabidopsis thaliana* is localized in the nucleus by a NLS in the C-terminal tail[30]. Compared to human, fly, and nematode, the lower eukaryotes including the *Plasmodium* and *Toxoplasma gondii*, have evolved EB1 proteins with a longer C-terminal extension, although the amino acid sequences are not conserved (Supplementary Fig. 5). Importantly, deletion of the genomic sequence encoding either the N-terminal tail, CH, or linker domain has no observable effect on the nuclear localization of PyEB1 (Fig. 5c, d). Therefore, the C-terminal tail (348-431 aa) of PyEB1 likely harbors a NLS that endows PyEB1 nuclear localizing.

Phylogenetic analyses using the amino acid sequences of EB1 proteins showed that apicomplexan protozoans, including *Cryptosporidium*, *Eimeria*, *Plasmodium*, *Toxoplasma gondii*, *Theileria*, and

*Babesia*, form a cluster that is evolutionarily far away from the other eukaryotic species (Supplementary Fig. 11). In agreement with this, TgEB1 has been observed to be targeted to the full-length MTs of the spindles at the tachyzoite stage[57]. A recent study also reports a co-localization of EB1 with nuclear MTs when episomally expressed in the gametocytes of *P. falciparum*[58]. These findings, together with our results, suggest that certain apicomplexan species may evolve EB1 protein possessing the MT-lattice affinity to fulfill the special requirement of MT cytoskeleton. In the future, it will be interesting to investigate the MT binding and localization of EB1 orthologous from the other apicomplexan protozoans.

Why is EB1 expressed and required in the male gametogenesis but not in the erythrocytic schizogony although the endomitotic spindles are analogously essential for both cell division processes? In erythrocytic schizogony, allocation of the replicated chromosomes and organelles into daughter cells are achieved by cytokinesis following the last round of endomitosis. The plasma and cortex membranes invaginate and engulf the cytosolic and nuclear contents, forming 16–28 daughter haploid merozoites[17,59]. However, following the last round of endomitosis in the male gametogenesis, one hemispindle presumably captures one haploid set of chromosomes and is egressed off from the nuclear envelope and then plasma membrane of mother gametocyte by axoneme-mediated exflagellation. Therefore, generation of daughter merozoite displays a mode of internal invagination inside the mother schizont, which is remarkably different from the external budding of daughter male gametes from mother gametocyte. Compared to the erythrocytic schizogony, the male gametogenesis may require higher velocity and stronger mechanical force for hemispindles to segregate chromosomes, because this process needs to be completed in a shorter time window. Consistent with this speculation, the umbrella-like array of hemispindles is defective after genetic disruption of EB1. PyEB1 also displays the stabilizing activity for the MTs in vitro. Therefore, the spindle MTs are decorated along their length with PyEB1 that could be responsible for their structure stability under high dynamic. It is noteworthy that the expression of EB1 is also detected in the oocysts (Fig. 1a). After EB1 disruption, the parasites fails to produce any oocysts or sporozoites in the mosquitoes (Fig. 1h, i), although considerable number of nucleated male gametes is observed (Fig. 2f), implying that EB1 is also critical for oocyst sporogony. In the future, it will be interesting to elucidate the function and mechanism of EB1 in the oocyst sporogony for daughter sporozoite formation.

In addition to the protective role in the spindle structure, PyEB1 was also critical in the spindle-kinetochore attachment during the male gametogenesis. In the canonical bi-orientational mitosis, chromosome segregation is achieved by the attachment of sister kinetochores to the plus-end of MTs, extending from the opposite spindle poles. Spindle MTs find the kinetochores of chromosomes via "search and capture"- a process that relies on the dynamic instability of MT plus-ends[11]. However, unbiased and stochastic search would theoretically require considerably more time for the MT plus-ends to capture all the kinetochores than the duration of mitosis based on mathematical modeling[60]. Notably, kinetochores could also be attached to the lateral surfaces of the bipolar spindle MTs[61–63]. This lateral attachment is likely complementary to the end-on attachment for efficient chromosome capture in a timely fashion. Different from the bi-orientational mitosis, the chromosome segregation of eight diploid genomes occurs simultaneously by eight hemispindles following three rounds of genome replication during *Plasmodium* male gametogenesis. However, the attachment status between hemispindles and kinetochores during this process remains unexplored due to the lack of specific marker for spindle as well as the resolution limitation of traditional confocal imaging. In this study, using PyEB1 as a spindle marker, we observe that the kinetochores are localized to the lateral side of hemispindle MTs throughout endomitosis. Consistent with it, several recent studies have also revealed similar MT lateral positioning of kinetochores in the

erythrocytic schizogony and male gametogenesis[9,64]. Interestingly, in the human cells with artificially induced monopolar spindle, which structurally mimics the hemispindles of *Plasmodium*, lateral attachment of kinetochores with MT is also detected as an intermediate status during the lateral to end-on attachment conversion[65]. These results suggest that spindle-kinetochore lateral attachment likely predominates over end-on attachment for chromosome capture or connection in *Plasmodium* endomitosis, which may increase the interface for hemispindle MT-kinetochore contact and occupancy.

Male gamete formation is often defective in chromosome segregation since about 20-30% of the WT male gametes lack detectable nuclei, as is observed in a previous study[12] and in this study (Fig. 2f). The assembly of hemispindles and the spindle-kinetochore attachment are both crucial for accurate and timely chromosome segregation in the male gametogenesis. The even binding of PyEB1 with full-length MTs in the hemispindles may behave like a quality control mechanism, facilitating the spindle-kinetochore lateral attachment in a direct or indirect manner (see the proposed working model in Supplementary Fig. 12). The discovery of PyEB1 with MT-lattice binding that contributes to the spindle-kinetochore lateral attachment provides insight into the adaptation of chromosome segregation in the *Plasmodium* male gametogenesis.

## Methods

### Mice and mosquitoes usage and ethics statement

All animal experiments were performed in accordance with approved protocols (XMULAC20140004) by the Committee for Care and Use of Laboratory Animals of Xiamen University. Female ICR mice (5–6 weeks old) were housed in Animal Care Center of Xiamen University (at 22–24 °C, relative humidity of 45–65%, a 12 h light/dark cycle) and used for parasite propagation, drug selection, parasite cloning, and mosquito feedings. Larval of the *Anopheles stephensi* mosquitoes (*Hor* strain) were reared at 28 °C, 80% relative humidity, and a 12-h light/12-h dark condition in a standard insect facility. Adult mosquitoes were supplemented with 10% (w/v) sugar solution and maintained at 23 °C.

### Plasmid construction and parasite transfection

The parasite CRISPR/Cas9 plasmid pYCm was used for gene editing[31]. To construct vectors for gene deletion, the 5′ and 3′ genomic fragments (400–800 bp) at the target gene were amplified as the left and right homologous templates respectively using the primers listed in Supplementary Table 1, and inserted into the pYCm vector. To construct vectors for gene tagging, the 5′- and 3′- flanking sequences (400–800 bp) at the designed insertion site of target genes were amplified as the left and right homologous templates respectively. DNA fragments encoding 6HA, 4Myc, GFP, and mScarlet were inserted between the homologous templates in frame with the coding sequence of target gene. For each modification, at least two small guide RNAs (sgRNAs) (Supplementary Table 1) were designed using the online program EuPaGDT. Paired oligonucleotides for sgRNA were denatured at 95 °C for 3 min, annealed at room temperature for 5 min, and ligated into pYCm. For parasite electroporation, parasite-infected red blood cells were electroporated with 5 μg plasmid DNA using Lonza Nucleofector. Transfected parasites were immediately intravenously injected into a naïve mouse and exposed to pyrimethamine (6 mg/mL) provided in mouse drinking water 24 h after injection.

### Genotyping of genetic modified parasites

All modified or transgenic parasites (listed in Supplementary Table 2) were generated from the *P. yoelii* 17XNL or 17XNL-derived parasite lines. In all, 10 μL parasite-infected blood samples were collected from the infected mice tail vein and red blood cells were lysed using 1% saponin in PBS. Parasite cells were spun down by centrifugation at 13,000 × *g* for 5 min and pellets were washed twice with PBS and boiled at 95 °C for 10 min followed by centrifugation at 13,000 × *g* for 5 min.

Supernatants containing parasite genomic DNA were subjected for genotyping. For each gene modification, both the 5′ and 3′ homologous recombination events were detected by diagnostic PCR, confirming successful integration of the homologous templates (Supplementary Fig. 13). Parasite clones with targeted modifications were obtained by limiting dilution. At least two clones of each gene-modified parasite were used for phenotypic analysis. Modified parasite clones subject for additional modification were negatively selected to remove pYCm plasmid. Each naïve mouse infected with the pYCm plasmid-carrying parasites was exposure to 5-Fluorouracil (5-FC, Sigma-Aldrich, cat#F6627) in mouse drinking water (2.0 mg/mL) for 6–8 days. After negative selection with 5-FC, two pairs of pYCm-specific primers are used for surveying residual plasmids in the parasites. All PCR primers used in this study are listed in the Supplementary Table 1.

### Parasite asexual blood stage proliferation in mouse

Parasite proliferation in the asexual blood stage was evaluated in mice. At least four ICR mice were included in each group. After intravenous injection of ~$1.0 \times 10^5$ parasites, parasite growth was monitored by Giemsa-stained thin blood smears every 2 days from day 2 to 14 post infection. The parasitemia is calculated as the ratio of parasitized erythrocytes over total erythrocytes.

### Gametocyte induction in mouse

ICR mice were treated with phenylhydrazine (80 mg/g mouse body weight) through intraperitoneal injection. Three days post injection, the mice were infected with $4.0 \times 10^6$ parasites through intravenous injection. Gametocytemia usually peaks at day 3 post infection. Male and female gametocytes were counted after Giemsa-stained thin blood smears. Sex-specific gametocytemia was calculated as the ratio of male or female gametocytes over parasite-infected red blood cells. All experiments were repeated three times independently.

### Gametocyte purification

Gametocytes were purified using the method described previously[66]. Mice were treated with phenylhydrazine 3 days before parasite infection. From day 3 post infection, infected mice were treated with sulfadiazine at 20 mg/L in drinking water to eliminate asexual blood stage parasites. After 48 h sulfadiazine treatment, mouse blood containing gametocytes was collected from orbital sinus into a heparin tube. Gametocytes were separated from the uninfected erythrocyte by centrifugation using 48% Nycodenz solution (27.6% w/v Nycodenz in 5 mM Tris-HCl, 3 mM KCl, 0.3 mM EDTA, pH 7.2,) and prepared in gametocyte maintenance buffer (GMB, 137 mM NaCl, 4 mM KCl, 1 mM CaCl2, 20 mM glucose, 20 mM HEPES, 4 mM NaHCO3, pH 7.24–7.29, 0.1% BSA). Gametocytes were harvested from the interphase and washed three times in the GMB buffer. All the operations were performed at 19–22 °C.

### Male gametocyte exflagellation assay in vitro

In all, 2.5 μL of mouse tail blood with 4–6% gametocytemia were added to 100 μL exflagellation medium (RPMI 1640 supplemented with 10% fetal calf serum and 50 μM XA, pH 7.4) containing 1 μL of 200 units/ml heparin. After 10 min of incubation at 22 °C, the numbers of ECs and RBCs were counted in a hemocytometer under a light microscope. The percentage of RBCs containing male gametocytes was calculated from Giemsa-stained smears, and the number of ECs per 100 male gametocytes was then calculated as exflagellation rate.

### In vitro ookinete differentiation

Mouse blood samples carrying 6–10% gametocytemia were collected and immediately added to ookinete culture medium (RPMI 1640, 10% FCS, 100 μM XA (Sigma-Aldrich, cat#D120804), 25 mM Hepes; pH 8.0, 100 μg/mL streptomycin, 100 U/mL penicillin). The gametocytes were

cultured at 22 °C for 12–15 h to allow gametogenesis, fertilization, and ookinete differentiation. Ookinetes formation was evaluated based on cell morphology in Giemsa-stained thin blood smears. Mature ookinete conversion rate was calculated as the number of crescent-shaped mature ookinete (stage V) over that of total ookinetes (from stage I to V).

## Parasite infection and transmission in mosquito

Thirty female *Anopheles stephensi* mosquitoes in one cage were allowed to feed on one anesthetized mouse carrying 6–10% gametocytemia for 30 min. For midgut oocyst counting, mosquito midguts were dissected on day 7 or 8 post blood feeding and stained with 0.1% mercurochrome for oocyst observation. For salivary gland sporozoite counting, mosquito salivary glands were dissected on day 14 post blood feeding, and the average number of sporozoites per mosquito was calculated.

## Chromosome segregation in the activated male gametocytes

To quantify the chromosomes allocation in the male gametocytes at 8 mpa, the gametocytes were fixed using 4% paraformaldehyde for 10 min, rinsed twice with PBS, and blocked with 5% BSA in PBS for 1 h. Parasites were then incubated with an antibody (Sigma-Aldrich, cat#T6199) recognizing both α-Tubulin I and α-Tubulin II proteins for 1 h and washed twice with PBS. Parasites were then incubated with fluorescent conjugated secondary antibodies for 1 h and followed by three washes with PBS. Parasites were then stained with DNA dye Hoechst 33342 (Thermo Fisher Scientific, cat#23491-52-3) for 10 min and mounted in a 90% glycerol solution. Male gametocytes (8 mpa) (Tubulin-positive) showing chromosome segregation were measured. Images were captured using identical settings under a ZEISS LSM 980 confocal microscope.

## DNA quantification in male gametes

To quantify the male gametes containing chromosomes at 15 min post activation (mpa), male gametes (15 mpa) were fixed using 4% paraformaldehyde for 10 min, rinsed twice with PBS, and blocked with 5% BSA in PBS for 1 h. Parasites were then incubated with an antibody (Sigma-Aldrich, cat#T6199) recognizing both α-Tubulin I and α-Tubulin II proteins for 1 h and washed twice with PBS. Parasites were then incubated with fluorescent conjugated secondary antibodies for 1 h and followed by three washes with PBS. Parasites were then stained with DNA dye Hoechst 33342 (Thermo Fisher Scientific, cat#23491-52-3) for 10 min and mounted in a 90% glycerol solution. Male gametes (Tubulin-positive) containing chromosomes (Hoechst 33342 -positive) were measured. Images were captured using identical settings under a ZEISS LSM 980 confocal microscope.

## Antibodies and antiserum

The primary antibodies used included: rabbit anti-HA (Cell Signaling Technology, cat#3724S, 1:1000 for immunoblotting IB, 1:1000 for immunofluorescence IF), rabbit anti-Myc (Cell Signaling Technology, cat#2272S, 1:1000 for IF, 1:1000 for IB). Mouse anti-Myc (Cell Signaling Technology, cat#2276S, 1:1000 for IF, 1:1000 for IB), mouse antibody against both α-Tubulin I and α-Tubulin II (Sigma-Aldrich, cat#T6199, 1:1000 for IF, 1:1000 for IB) for staining the MT in the parasites, mouse anti-β-Tubulin (Sigma-Aldrich, cat#T5201, 1:1000 for IF, 1:1000 for IB) for staining the MT in the parasites, rabbit anti-Histone H3 (Abcam, cat#ab1791, 1:1000 for IB), rabbit anti-Polyglutamate chain (PolyE) (AdipoGen, cat#AG-25B-0030, 1:1000 for IF), mouse monoclonal antibody EB1 (BD Biosciences, cat#610535, 1:300 for IF), rabbit monoclonal antibody against α-tubulin YL1/2 (Thermo Fisher Scientific, cat#MA1-80017, 1:1000 for IF) for staining the MT in mammalian cells. The secondary antibodies used were as follows: HRP-conjugated goat anti-rabbit IgG (Abcam, cat#ab6721, 1:5000 for IB), HRP-conjugated goat anti-mouse IgG (Abcam, cat#ab6789, 1:5000 for IB),

Alexa 555 goat anti-rabbit IgG (ThermoFisher Scientific, cat#A21428, 1:1000 for IF), Alexa 488 goat anti-rabbit IgG (ThermoFisher Scientific, cat# A31566, 1:1000 for IF), Alexa 555 goat anti-mouse IgG (ThermoFisher Scientific, cat#A21422, 1:1000 for IF), Alexa 488 goat anti-mouse IgG (ThermoFisher Scientific, cat#A11001, 1:1000 for IF), Alexa 594 goat anti-mouse IgG (ThermoFisher Scientific, cat#A11032, 1:300 for IF), and Alexa 594 goat anti-rat IgG (ThermoFisher Scientific, cat#A-11007, 1:300 for IF). The anti-serums included rabbit anti-P28 (our lab, 1:1000 for IFA), rabbit anti-BiP (our lab, 1:1000 for IB), and rabbit anti-EB1pSer15 (our lab, 1:200 for IB). The designed phosphorylated polypeptide (CSFGNMDSGFVV) of EB1 was conjugated with KLH for immunization, mixed with Freund's adjuvant, and immunized in the New Zealand white rabbits. After three subcutaneous immunizations, whole blood of rabbits was taken, and the supernatant was collected after centrifugation.

## Live cell imaging

Parasites expressing GFP-fused proteins were collected in 200 μL PBS, washed twice with PBS and stained with Hoechst 33342 at room temperature for 10 min. After centrifugation at 500 g for 3 min, the parasites pellets were re-suspended in 100 μL of 3% low melting agarose (Sigma-Aldrich, A9414), and transferred evenly on the bottom of a 35-mm culture dish. Parasites were placed at room temperature for 15 min and imaged using a GE OMX V4 Ultra-high resolution microscope with the 60 × 1.7 oil objective.

## Immunofluorescence assay

Purified parasites were fixed with of 4% paraformaldehyde for 15 min and rinsed with PBS three times. The parasites were then permeabilized with 0.1% Triton X-100 for 10 min, rinsed with PBS twice, and incubated with 5% BSA for 1 h. They were incubated with the primary antibodies diluted in 3% BSA-PBS at 4 °C for 12 h, rinsed with PBS three times, and incubated with fluorescent conjugated secondary antibodies for 1 h in the dark. After three washes with PBS, they were stained with DNA dye Hoechst 33342 for 10 min and mounted on glass slides using mounting medium (90% glycerol). All images were acquired and processed using identical settings on Zeiss LSM 880 and LSM 980 confocal microscopes.

## Ultrastructure expansion microscopy (U-ExM)

Purified gametocytes were sedimented on a 15 mm round poly-D-lysine (Sigma-Aldrich, cat#A-003-M) coated coverslips for 10 min. The parasites were then permeabilized with 100% ice-cold methanol for 7 min. To add anchors to proteins, coverslips were incubated for 5 h in 1.4% formaldehyde (FA, Sigma-Aldrich, cat#F8775)/2% acrylamide (AA, Sigma-Aldrich, cat#146072) at 37 °C. Next, gelation was performed in ammonium persulfate (APS, Sigma-Aldrich, cat#A7460))/ *N,N,N′,N′*-Tetramethyl ethylenediamine (TEMED, Sigma-Aldrich, cat#110-18-9)/Monomer solution (19% Sodium Acrylate (SA, Sigma-Aldrich, cat#408220); 10% AA; 0.1% *N,N′*-Methylene-bisacrylamide (BIS-AA, Sigma-Aldrich, cat#M7279) in PBS) for 1 h at 37 °C. Sample denaturation was performed for 60 min at 95 °C. Gels were incubated in bulk ddH$_2$O at room temperature overnight for complete expansion. The following day, gel samples were washed in PBS twice for 30 min each to remove excess ddH$_2$O. Gels were then cut into square pieces (-1 cm × 1 cm), incubated with primary antibodies at 37 °C for 3 h, and washed with 0.1% PBS-Tween (PBS-T) 3 times for 10 min each. Incubation with the secondary antibodies was performed for 3 h at 37 °C followed by 3 times washes with 0.1% PBS-T 10 min each. In some conditions if needed, gels were additionally stained by NHS-ester (Merck, cat#08741) diluted at 10 μg/mL in PBS for 90 min at room temperature. After the final staining step, gels were then washed with 0.1% PBS-T 3 times for 15 min each and expanded overnight by incubating in bulk ddH$_2$O at room temperature. After the second round of expansion, gels were cut into

square pieces (~0.5 cm × 0.5 cm) and mounted by a coverslip in a fixed position for image acquiring.

## Protein extraction and immunoblot
Protein extracts from the asexual blood stage parasites, gametocytes, and ookinetes were lysed in RIPA buffer (0.1% SDS, 1 mM DTT, 50 mM NaCl, 20 mM Tris-HCl; pH 8.0) (Solaribio, cat#R0010) supplemented with protease inhibitor cocktail (Medchem Express, cat#HY-K0010) and PMSF (Roche, cat#10837091001). After ultrasonication, the extracts were incubated on ice for 30 min followed with centrifugation at 12,000 × $g$ for 10 min at 4 °C. Clarified supernatant was mixed with same volume of 2x protein sample buffer, boiled at 95 °C for 5 min, and cooled at room temperature. After SDS-PAGE separation, samples were transferred to PVDF membrane (Millipore, cat#IPVH00010). The membrane was blocked with 5% skim milk, probed with primary antibodies for 1 h at room temperature, rinsed 3 times with TBST, and incubated with HRP-conjugated secondary antibodies. Followed by three washes with TBST, the membrane was visualized with enhanced chemiluminescence detection (Advansta, cat#K12045-D10).

## Protein immunoprecipitation
Parasites were lysed in IP buffer A (50 mM HEPES pH 7.5, 150 mM NaCl, 1 mM EDTA, 1 mM EGTA, 1% Triton X-100, 0.1% sodium deoxycholate) with protease inhibitor cocktail and PMSF. Protein aggregates were pre-cleared by centrifugation at 20,000 × $g$ for 10 min, and 1 mL of lysates were incubated with primary beads (nanobody-HA or nanobody anti-Myc) mixed for 3 h. The beads were washed with IP buffer A for three times at 4 °C, and then mixed with an equal volume of 2×SDS sample buffer for protein elution. All samples were boiled at 95 °C for 10 min and centrifuged at 12,000 × $g$ for 5 min. Equal volume of supernatants from each sample were used for immunoblotting.

## Flow cytometry analysis
The infected mouse blood containing gametocytes was collected after induction by phenylhydrazine. After two washes with PBS, the cells were suspended in PBS with Hoechst 33342 (Thermo Fisher Scientific, cat#23491-52-3) for nuclei staining and analyzed by flow cytometry using BD LSR Fortessa flow cytometer. Parasite-iRBCs were first gated using the fluorescence signal of 405 nm (Hoechst 33342). GFP-positive male gametocytes and mCherry-positive female gametocytes were gated using 488 and 561 nm, respectively. All the data were analyzed using FlowJo.

## Transmission electron microscopy
Purified gametocytes (8 mpa) were fixed with 2.5% glutaraldehyde in 0.1 M phosphate buffer at 4 °C overnight, rinsed three times with 0.1 M phosphate buffer, post-fixed with 1% osmium acid for 2 h, and rinsed three times with 0.1 M phosphate buffer and dehydrated with concentration gradient ethanol. After embedding and slicing, thin sections were stained with uranyl acetate and lead citrate before imaging. All samples were imaged under the HT-7800 electron microscope.

## Mammalian cell culture, transfection, and protein purification
HEK293T cells and MRC5 cells were cultured in DMEM/F12 (Biosharp, cat#BL305A) supplemented with 10% FBS, 100 U/mL penicillin, and 100 μg/mL streptomycin and kept at 37 °C in 5% CO2. FuGENE6 (Promega, E2691) was used to transfect plasmids into MRC5 cells for immunofluorescence assay. Polyethylenimine (PEI; Polysciences) was used to transfect plasmids into HEK293T cells for protein purification. For protein purification in HEK293T cells, pTT5 vector was used for expression of HsEB1 or PyEB1 fragments. HEK293T cells grown on 15-cm dishes were transfected with 30 μg DNA per dish, cells were treated with 1 μM nocodazole overnight before harvesting. 36 h post transfection, the transfected cells were collected with cold PBS (4 °C, 10 ml for each 15-cm dish) into 15-mL falcon tubes. Cells were

centrifuged at 300 g, 4 °C for 10 min to remove the supernatant, and lysed in 900 μL lysis buffer (50 mM Hepes, 300 mM NaCl, and 0.5% Triton X-100, pH 7.4) containing protease inhibitors (Roche) for 10 min on ice. Cell lysate was centrifuged at 14,000 × $g$, 4 °C for 20 min, and the supernatant was incubated with 40 μL StrepTactin beads (GE Healthcare, cat#28-9356-00) at 4 °C for 45 min. After removal of the supernatant by centrifuging at 500 × $g$, 4 °C for 1 min, beads were washed four times with 1 mL lysis buffer and twice with 1 mL wash buffer A (50 mM Hepes, 150 mM NaCl, and 0.01% Triton X, pH 7.4). Proteins were eluted with 40 μL elution buffer (50 mM HEPES, 300 mM NaCl, 0.05% Triton X-100, 10% glycerol, and 2.5 mM desthiobiotin, pH 7.4), snaped frozen, and stored at −80 °C.

## Preparation of GMPCPP-stabilized microtubule seeds
Double-cycled GMPCPP MT seeds were made as described previously[67]. Briefly, 8.25 μL tubulin reaction mixture in MRB80 buffer (80 mM PIPES, 1 mM EGTA, and 4 mM MgCl2, pH 6.8), which contained 14 μM unlabeled porcine brain tubulin (Cytoskeleton, cat#T238P-C), 3.6 μM biotin-tubulin (Cytoskeleton, cat#T333P), 2.4 μM rhodamine-tubulin (Cytoskeleton, cat#TL590M), and 1 mM GMPCPP (Jena Biosciences, cat#NU-405L), was incubated at 37 °C for 30 min. MTs were then pelleted by centrifugation in an Airfuge (Beckman) at 28 psi (pounds per square inch) for 5 min. Followed by carefully removing the supernatant, the pellet was resuspended in 6 μL MRB80 buffer and depolymerized on ice for 20 min. Subsequently, a second round of polymerization was performed at 37 °C in the presence of freshly supplemented 1 mM GMPCPP for 30 min. The MT seeds were then pelleted as described above and resuspended in 50 μl MRB80 buffer containing 10% glycerol, snap frozen, and stored at −80 °C.

## TIRF assays, image acquisition, and data processing
Flow chambers used for assays were made of the plasma-cleaned glass coverslips and microscope slides. The flow chambers were sequentially incubated with 0.2 mg/mL Poly-ʟ-Lysine-polyethylene glycol (PLL-PEG)-biotin (Susos AG, Switzerland) and 1 mg/mL neutravidin (Invitrogen, cat#31000) in MRB80 buffer. GMPCPP seeds were then attached to coverslip via biotin-neutravidin links, followed by blocking with 1 mg/ml κ-casein (Sigma, C0406-1). The reaction mixture, which consisted of purified protein and MRB80 buffer containing 20 μM porcine brain tubulin, 0.5 μM rhodamine-tubulin, 1 mM GTP, 0.2 mg/ml κ-casein, 0.1% methylcellulose, and oxygen scavenger mix (50 mM glucose, 400 μg/ml glucose oxidase, 200 μg/mL catalase, and 4 mM DTT), was added to the chamber after centrifugation with Airfuge for 5 min at 28 psi. The flow chamber was sealed with vacuum grease and imaged immediately at 30 °C using a TIRF microscope (Nikon Eclipse Ti2-E). The imaging interval was 2 s unless otherwise stated.

TIRF microscopy was performed on Nikon Eclipse Ti2-E with the perfect focus with the Nikon CFI Apo TIRF 100 × 1.49 NA oil objective, Prime 95B camera (Photometrics), SOLE laser engine (four lasers: 405 nm, 488 nm, 561 nm, and 638 nm; Omicron) and controlled by NIS-Elements software (Nikon). Images were magnified with a 1.5× intermediate lens on Ti2-E before projected onto the camera. The resulting pixel size is 73.3 nm/pixel. Stage top incubator INUBG2E-ZILCS (Tokai Hit) was used to keep samples at 30 °C in vitro. Microtubule dynamics were measured by producing kymographs using the Multi Kymograph function of the Fiji image analysis software and manually fitting lines to growth and shrinkage events. Catastrophe frequencies were calculated as the number of events divided by total microtubule growth time. Rescue frequencies were calculated as the number of rescue events divided by total microtubule depolymerization time; events without a rescue were assigned a value of 0.

## Bioinformatic searches and tools
The genomic sequences of target genes were downloaded from Plas-moDB database (http://plasmodb.org/plasmo/app/)[68]. The sgRNA of

target genes were searched using database EuPaGDT (http://grna.ctegd.uga.edu/)[69]. The codon usage was optimized using JCat (http://www.prodoric.de/JCat)[70]. The protein homologs of EB1 from species in different taxonomies were identified with the hidden Markov model-based method HMMER. The amino acid sequences of EB1 homologs were downloaded from UNIPROT (https://www.uniprot.org/) and aligned with MUSCLE (Version 5.1). Aligned sequences were trimmed with TrimAl (Version 1.4.1), and phylogenetic tree was generated with IQ-Tree (Version 2.2.0). Confidence level of the phylogenetic tree was tested with IQ-Tree ultrafast bootstrap.

## Quantification and statistical analysis

For quantification of protein relative expression level in immunoblot, protein band intensities were quantified using by Fiji software[71]. For quantification of the length, angle and volume of hemispindles in confocal fluorescence microscopy, images was quantified using Fiji software[71]. The 3D surface topology reconstruction of Z-stack images was quantified by Imaris X64 9.2.0. For quantification of protein expression in IFA, images were acquired under identical parameters. Fluorescent intensities of proteins were quantified using ZEISS software. More than 50 cells were randomly chosen in each group. All graph-making and statistical analysis were performed using Graphpad Prism 8.0. Data collected as raw values are shown as mean ± SEM or mean ± SD or means only if not otherwise stated. Details of statistical methods are reported in the figure legends. Two-tailed Student's $t$ test or Mann–Whitney $U$ test was used to compare differences between control and experimental groups. Statistical significance is shown as $*p < 0.05$, $**p < 0.01$, $***p < 0.001$, $****p < 0.0001$, ns, not significant. n represents the sample volume in each group or the number of biological replicates.

## Reporting summary

Further information on research design is available in the Nature Portfolio Reporting Summary linked to this article.

## Data availability

All relevant data in this study are submitted as supplementary source files. Source data are provided with this paper.

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

## Acknowledgements

We thank Prof. Yiming Zheng (Xiamen University), Prof. Rui Zhang (Washington University), and Prof. Xin Liang (Tsinghua University) for the comments on this manuscript. This work was supported by the National Natural Science Foundation of China (32270503, 32200554, 32170427, 32000445, 31970387, and 32070705), the Natural Science Foundation of Fujian Province (2021J01028), and the 111 Project sponsored by the State Bureau of Foreign Experts and Ministry of Education of China (BP2018017).

## Author contributions

S.Y., M.C., and S.Z. generated the modified parasites, conducted the phenotype analysis, IFA assay, image analysis, mosquito experiments, J.H. performed the in vitro and in vivo MT-binding experiments, X.M. performed the bioinformatics analysis, J.Y., H.C., and K.J. supervised the work, S.Y., M.C., J.H., H.C., K.J., and J.Y. analyzed the data, and J.Y. wrote the manuscript.

## Competing interests

The authors declare no competing interests.
