## [Peer Review File · Nature Communications]

Reviewer comments, first round review

Reviewer #1 (Remarks to the Author):

Review of NCOMM-23-04049
Yang, et al (Yuan)

The manuscript by Yang et al. investigates the role of EB1 during male gamete formation in Plasmodium parasites. The connections between the spindle and kinetochores are essential and have not been extensively studied in Plasmodium. In the current manuscript the authors demonstrate that EB1 has unexpected features, including loss of plus-end binding to microtubules. EB1 association with microtubules instead occurs throughout the "microtubule lattice". This manuscript utilizes the rodent parasite, Plasmodium yoelii, as the major model. The authors use high-quality genetics for both knockout and complementation studies, adding to the robustness of the results.

The study provides an interesting and in-depth analysis of EB1. The findings will be important for the parasitology and larger cell biology communities.

Major comments:

1. The grammar and language selection have multiple minor errors throughout. It would be helpful to have some help with proof-reading and editing. A few examples are included below.
2. On figure 1J, K, L: the authors should use more standard genetic notation for their complementations. If this is not possible, then refer to the parasites with something that includes the parasite source for the complementation (Py vs Pf).
3. For measurements that are made with multiple biological replicates, the measurement should be provided with an error or standard deviation or range. For example, in line 176, are all 60 samples counted in a single biological replicate? If so, these should be repeated. If not, then the measurement should be 49% +/- standard deviation (or something similar). This is true throughout the manuscript.
4. Figure 3A: only 7 spots of EB are visible in the 8-9 mpa image. A more representative image should be shown. Legend for 3F does not include a description of all of the colors. The dashed line is presumably the nuclear membrane, but this is not described.
5. In 4B-1: it looks like GFP-HsEB1 also shows some staining along the length of the microtubules. Is this expected and real or an artifact from overexpression; because 4C looks much better.
6. In figures 4D, why is there a central (red) region where PyEB1 does not bind or does not bind as efficiently? It would be reasonable to comment on this in the text. Similar comment for HsEB1 and PyEB1 in 4F – what is the central strip of dual binding?
7. Line 344 – more precise language should be used to describe the defect. It is true that spindle assembly seems intact for the deltaEB1 parasites, but it is unclear what is meant by "spindle structure".
8. The data in figure 7B are very interesting and present a shift in the view of these attachments. This should be validated to rule out technical reasons for the finding. The fluorophores on the secondary antibodies should be swapped to make sure the localization still holds by U-ExM. In other words, the green and red channels should be swapped (so secondary for the HA primary is green; secondary for myc primary is red in one set. In a second set, the secondary for the HA primary should be shifted to the red channel and the secondary for the myc primary shifted to the green channel). This is control for possible chromatic aberration of the microscope objective.
9. The conclusion in lines 451-454 is not supported by the data. The phosphorylation of S15 may be important or allows efficient interaction of EB1 with Ndc80, but it cannot be said to be critical. In the S15A mutant 32% of Ndc80 was still spindle-attached.

Minor comments:

1. Line 121 – affirmed should be confirmed; line 173 – individual "chromosome is" should be

“chromosomes are”; line 216 – at spindle should be at the spindle; line 304 – remarked should be remarkable or major or notable.

2. Line 127 – a citation should be provided for the male gametocyte marker. It looks like the antibody that was used is not predicted to distinguish alpha tubulin I from II. Is alpha-tubulin II is the male-specific marker, or is it generic alpha tubulin? This may be know but is not discussed in the text.

Reviewer #2 (Remarks to the Author):

This manuscript by Yang et al describes work investigating the role of the microtubule-binding protein, EB1, in male gametogenesis in *Plasmodium yoelii*.

The authors show that PyEB1 is not expressed in asexual blood stage parasites and that knockout of PyEB1 did not affect asexual blood stage growth of gametocytogenesis.

PyEB1 is present from the initiation of gametocytogenesis, but more abundant in male gametocytes. Immunofluorescence microscopy, including Expansion Microscopy, and live cell imaging of EB1-GFP transfectants, following activation of male gametocytes yielded images consistent with EB1 association with the spindle.

Fig S2B. The diagram shows the SAS4 (basal body protein) as located inside the nucleus. Presumably this should always be in the cytoplasm.

Knockout of PyEB1 has no effect on female gamete formation; but leads to a decrease in the number of exflagellation centres. The failure to form midgut oocysts is therefore due to a male gamete defect. The defect is associated with failure to correctly segregate the DNA. The defect could be complemented with either PyEB1 or PfEB1, suggesting functional conservation.

In a very nice heterologous expression experiment, the authors transfected GFP-tagged PyEB1 and HsEB1 into MRC5 cells. They found that HsEB1 bound to the ends of spindle microtubules as expected, while PyEB1 bound along the MTs. They also purified recombinant GFP-tagged PyEB1 and HsEB1 and showed that PyEB1 seeds MT lattices and has MT stabilising activity, while HsEB1 tracks the plus end. The authors dissected the regions responsible for binding and found that both the CH domain and the linker have MT binding ability. They found that the N-terminal region is not required for MT binding but does play a role in chromosome segregation.

Immunoprecipitation of EB1 revealed a potential interaction with NDC80 in activated male gametocytes. Given that EB1 is bound to microtubules, the immunoprecipitation protocol is likely to bring down microtubule fragments and a number of associated proteins. Thus, immunoprecipitation does not provide sufficient evidence to assume a direct interaction. However, I note that the authors are careful to conclude that “..EB1 regulates spindle-kinetochore attachment..” rather than concluding a direct interaction.

The authors have generated a very nice conditional knockdown system to look at the effects of NDC80 in male gametes. Not surprisingly, the data suggest that NDC80 is needed for kinetochore attachment.

The authors have also shown a critical role for EB1 Ser-15 phosphorylation for EB1 function in recruiting kinetochores to the spindle.

In the first part of the Discussion, the authors present the finding of PyEB1 binding along the length of the MTs as novel and surprising. They later point to recent studies in *Pf* (Ref 31) and *Toxoplasma gondii* (doi: 10.1091/mbc.E15-06-0437) showing the EB1 binds along the spindle MTs. Yeast EB1 has also been shown to bind along the MT lattice (DOI: 10.1038/ncomms11665). It might be better to nuance the Discussion to make it clear that the difference is with mammalian EB1, while the apicomplexan mitotic apparatus is more similar to that of yeast.

In conclusion, this is a very well-executed study that provides interesting insights into a potential role for EB1 in controlling spindle assembly and kinetochore attachment during male gametogenesis.

Minor points

Line 19. "Plasmodium evolves the endomitosis for this multinucleated cell division". It may be better to say "Plasmodium undergoes endomitosis for this multinucleated cell division." Endomitosis is a feature of all apicomplexans; and other organisms, such as yeast.

Line 23. Plasmodium EB1 is a unique orthologue. It is preferable not to use the term unique. As presented in Fig S11, Apicomplexan parasites likely have a functionally well-conserved EB1.

Line 61. "...A bipartite microtubule (MT) organization center (MTOC) [9], embedded in the nuclear envelope...". While it is bipartite, only one part is embedded in the nuclear envelope.

Reviewer #3 (Remarks to the Author):

The study addresses the temporal stage-specific expression, localization and function of the EB1 protein in *Plasmodium yoelii*. The authors demonstrate that EB1 is dispensable for asexual and pre-sexual stages, but required for finalizing sexual differentiation and oocyst formation, a critical step in malaria transmission. The authors show, by complementation experiments, that restituting EB1, of either *P. yoelii* or *P. falciparum*, is sufficient for parasites to complete their life cycle in mosquitoes. They further analyze the basis of the phenotype, and decipher that EB1 is required for male gamete flagellation. The authors specifically identify that genome segregation is impaired during male gamete formation, while other male gamete structure assemble correctly. Furthermore, using a heterologous expression system the authors identify that *Plasmodium*'s EB1 does not accumulate at MT's plus-end (as does its human counterpart), but rather binds MTs evenly, deciphering a fundamental mechanistic difference between the two proteins. Finally, the authors identify that spindle MT-binding is given by EB1's CH and linker domains. The authors also map a specific phosphorylation required for EB1's localization to the spindle. The authors unravel that the MT lattice binding capacity of EB1 is encompassed by a distinct distribution of kinetochore proteins along spindle MTs, with EB1 playing a critical role in kinetochore clustering.

I would like to commend the authors for the truly spectacular images, for their remarkable attention to detail, and for thoroughly exploring multiple plausible hypotheses throughout the manuscript using an impressive array of approaches. The quality of the manuscript is very good, and overall, the writing is clear and concise.

However, there are a few points that need addressing before the manuscript is in its final form for publication:

Line 103-104: the authors mention that "we reported that *Plasmodium* EB1 is a unique orthologue distinct from the eukaryotic EB1." What do the authors mean? *Plasmodium* is an eukaryote, hence *Plasmodium*'s EB1 is by definition an eukaryotic EB1. Did they mean human EB1? A similar statement is repeated in Lines 249-250

In Figure 1B a western blot is shown in which asexual blood stages and sexual stages are labeled. However, there are no markers for the stages that could confirm the majority of the proteome assayed for EB1 expression indeed corresponds to the indicated stages.

Line 128. It is stated that EB1 is more highly expressed in male gametocytes. The evidence provided for this statement is a single image in Figure 1C, which appreciably shows a higher fluorescence intensity signal for EB1 staining. However, there is no quantification of the signal, nor additional images for the reader to appreciate whether the images shown are representative of the variability between gametocyte types.

Figures 1L and 1J show that the Δ EB1 strain generates oocysts in 7-10% of mosquitoes. This is neither mentioned nor the implication discussed in the manuscript. Are there alternative or redundant mechanisms? Could this be an indication of parthenogenesis? Are the oocysts formed viable (i.e. do they contain viable sporozoites)? Please address this point.

Figure 1J shows that the Comp1 strain expresses a higher level of EB1 despite it being under its endogenous promoter. Please address the origin of this difference in your results' description, and how this could account for the slightly better complementation of the WT phenotype for Comp1 vs. Comp2 (results shown in Fig 1L). Could there be an effect of the expression level? Is *P. yoelii* better complemented by the *P. yoelii* protein than by the *P. falciparum* one? Please address this point

In general, there is reference in the figure legends to how many experiments were done per quantification. However, the sample size analyzed in each case is not mentioned. For example, In Figures 1F and G, the number of animals used in the quantifications is not specified. For figure 2A, how many individuals did you quantify per experimental replicate? For the results shown in Figure 1G, please provide the p-values obtained for both females and male gametes between experimental conditions.

Line 179. The authors report that 67% of the WT gametes are nucleated. Please comment on whether obtaining a 33% anucleate male gametes is within the expected range and provide a citation.

In figure 3C, anti-tubulin labels multiple structures. It would be helpful for the non-expert reader to point out the mitotic spindle clearly in these images, or to provide insets where the overlap between the tubulin signal at the spindle and EB1 can be more clearly appreciated.

While it is well appreciated that the authors were thorough in showing that the localization of EB1 during gametogenesis is reproducible despite the tag or antibody used, I envision some of the redundant panels which serve as validation for the different strains could go onto supplementary figures. On the other hand, Figure 4S contains important topological information about EB1's with respect to other structures during male gametogenesis which are important to comprehend its function. This is important information which in my opinion should be part of a main figure.

Minor suggestions:

The use of the English language should be thoroughly revised to ensure the reader's correct interpretation of the manuscript.

To point out a few examples (please note this is definitely not a comprehensive list):

1. Plasmodium is frequently referred to as "the Plasmodium" while every mention to the mitotic spindle is missing the required article "the."
2. There is a mix of verbal tenses used throughout the manuscript that should be revised. Please refer to your current findings, reported for the first time in this study, in the present tense (For example, in lines 103-104)
3. Please correct all "Plasmodium EB1" to "Plasmodium's EB1"
4. Either refer to "the Δ EB1 strain" or simply refer to it as " Δ EB1". Likewise, please refer to the EB1 protein, as either "the EB1 protein" or simply "EB1" (not "the EB1" alone).
5. Lines 206-207. "EB1 was dispersed at nucleoplasm while Tubulin was dominantly at cytoplasm" please rephrase for clarity
6. Lines 215-216, please rid of "the" in "we performed the Ultrastructure Expansion Microscopy," and add "the" to "... fine localization of EB1 at the spindle"

Line 109. "the spindle-kinetochore attachment was impaired for chromosome segregation in the male gametogenesis" ?? This sentence is unclear. Please rephrase for clarity.

Line 127. Please provide a reference for α -Tubulin as a marker for male gametocytes

Line 140. Please spell out *Anopheles stephensi*. To abbreviate, please refer to the species using

appropriate nomenclature (*A. stephensi*)

Line 149: please replace "was" with "were"

Line 540: please add "set" or "complement" between "haploid" and "of chromosomes".

Response to Reviewer Comments on the manuscript [NCOMMS-23-04049]:

Reviewer #1

The manuscript by Yang et al. investigates the role of EB1 during male gamete formation in Plasmodium parasites. The connections between the spindle and kinetochores are essential and have not been extensively studied in Plasmodium. In the current manuscript the authors demonstrate that EB1 has unexpected features, including loss of plus-end binding to microtubules. EB1 association with microtubules instead occurs throughout the “microtubule lattice”. This manuscript utilizes the rodent parasite, Plasmodium yoelii, as the major model. The authors use high-quality genetics for both knockout and complementation studies, adding to the robustness of the results.

The study provides an interesting and in-depth analysis of EB1. The findings will be important for the parasitology and larger cell biology communities.

Response: We thank the reviewer for the encouraging comments on our work.

Major comments:

1. The grammar and language selection have multiple minor errors throughout. It would be helpful to have some help with proof-reading and editing. A few examples are included below.

Response: Thank reviewer for pointing it out. We have a professional check the grammar and language throughout the manuscript.

2. On figure 1J, K, L: the authors should use more standard genetic notation for their complementations. If this is not possible, then refer to the parasites with something that includes the parasite source for the complementation (Py vs Pf).

Response:

In the Figure 1E, we provided a schematic which describes the detailed information (including the parasite source of protein rescue) of the modified parasite strains after *eb1* gene deletion and complementation, respectively. Meanwhile, these information were also provided in the legend of Figure 1.

3. For measurements that are made with multiple biological replicates, the measurement should be provided with an error or standard deviation or range. For example, in line 176, are all 60 samples counted in a single biological replicate? If so, these should be repeated. If not, then the measurement should be 49% +/- standard deviation (or something similar). This is true throughout the manuscript.

Response: Thank reviewer for pointing it out. We have added the standard deviation to all the measurements in the results throughout the revised manuscript.

4. Figure 3A: only 7 spots of EB are visible in the 8-9 mpa image. A more representative image should be shown.

Response: In these experiments, 6-7 spots of EB1 were most frequently observed in the gametocytes at 8-9 mpa while the gametocyte with 8 spots of EB1 was relatively less observed. This could be due to the different efficiency for the antibody staining in IFA, or due to the different position which is not captured by the confocal microscopy.

According to the reviewer's suggestion, we replaced a new cell containing 8 spots of EB in the 8-9 mpa image in the Figure 3A.

Legend for 3F does not include a description of all of the colors. The dashed line is presumably the nuclear membrane, but this is not described.

Response: Thank reviewer for pointing it out. We have added the description for all of the color and the dashed line in the revised legend of Figure 3F.

5. In 4B-1: it looks like GFP-HsEB1 also shows some staining along the length of the microtubules. Is this expected and real or an artifact from overexpression; because 4C looks much better.

Response: The reviewer is correct. In Figure 4B-1, GFP-HsEB1 also binds along the length of the MTs. This is expected because exogenous overexpression of HsEB1

could enhance its affinity for MT lattice, in agreement with previous results (PMID:12058019, 10899006, 16148041, and 23251535). However, in contrast to PyEB1, which binds evenly along the whole length of MTs, HsEB1 shows a strong preference for plus ends, as the intensity of HsEB1 at the plus end is much higher than that along the lattice.

In Figure 4C, the endogenous HsEB1 looks like binding exclusively at the plus ends. However, we should mention that weak lattice staining of endogenous HsEB1 could also be observed in some cells. Despite this, the notion that HsEB1 shows a strong preference for MT plus ends still stands.

6. In figures 4D, why is there a central (red) region where PyEB1 does not bind or does not bind as efficiently? It would be reasonable to comment on this in the text. Similar comment for HsEB1 and PyEB1 in 4F – what is the central strip of dual binding?

Response: In Figure 4D, the central red stripes (central strip, indicated as a white arrow in the below picture which is not included in the manuscript) are stable GMPCPP MT seeds used as templates to generate dynamic GDP MTs. Since the GMPCPP MT seeds contained 12% rhodamine-tubulin dimers while the newly generated GDP MTs contained only 2.5% rhodamine-tubulin dimers, the former looks brighter than the latter. The detailed information in this protocol were provided in the Material and Method. Judging from the “EB1 60 nM” images shown in the upper panels of Figure 4D, PyEB1 binds to the GMPCPP MT seeds as efficiently as to the newly generated GDP MTs. In the central seeds regions in merged images and kymographs, the red channel is visually dominant over the green channel, leading to the reviewer’s misunderstanding that PyEB1 does not bind efficiently to the central

regions.

In Figure 4F, both HsEB1-mCherry and GMPCPP microtubule seeds were shown in the red channel. Since HsEB1 does not bind to GMPCPP microtubule seeds, as shown in Figure 4D, the central red stripe signal comes from GMPCPP microtubule seeds, and the red tip signal comes from HsEB1-mCherry.

7. Line 344 – more precise language should be used to describe the defect. It is true that spindle assembly seems intact for the deltaEB1 parasites, but it is unclear what is meant by “spindle structure”.

Response: We changed “EB1 is dispensable for early spindle assembly but is important for the spindle structure” to “EB1 is dispensable for early spindle assembly but is important for the structure of later stage hemispindles” inline 341.

8. The data in figure 7B are very interesting and present a shift in the view of these attachments. This should be validated to rule out technical reasons for the finding. The fluorophores on the secondary antibodies should be swapped to make sure the localization still holds by U-ExM. In other words, the green and red channels should be swapped (so secondary for the HA primary is green; secondary for myc primary is red in one set. In a second set, the secondary for the HA primary should be shifted to the red channel and the secondary for the myc primary shifted to the green channel). This is control for possible chromatic aberration of the microscope objective.

Response: We observed the representative localization of EB1 and Ndc80 in Figure 7B from several dependent experiment repeats. To further address the reviewer’s concern, we repeated the experiment using the swapped secondary antibodies as the reviewer suggested. The results showed similar relative localization of EB1 and Ndc80 as in Figure 7B (see the below picture, not included in the revised manuscript).

9. The conclusion in lines 451-454 is not supported by the data. The phosphorylation of S15 may be important or allows efficient interaction of EB1 with Ndc80, but it cannot be said to be critical. In the S15A mutant 32% of Ndc80 was still spindle-attached.

Response: We changed “These results indicated that S15 phosphorylation is critical for EB1 function in the spindle-kinetochore attachment” to “These results indicated that S15 phosphorylation is important for EB1 function in the spindle-kinetochore attachment” in line 446.

Minor comments:

1. Line 121 – affirmed should be confirmed; line 173 – individual “chromosome is” should be “chromosomes are”; line 216 – at spindle should be at the spindle; line 304 – remarked should be remarkable or major or notable.

Response: Thank reviewer for pointing these errors out. We have changed accordingly.

2. Line 127 – a citation should be provided for the male gametocyte marker. It looks like the antibody that was used is not predicted to distinguish alpha tubulin I from II. Is alpha-tubulin II is the male-specific marker, or is it generic alpha tubulin? This may be know but is not discussed in the text.

Response:

In line 126, we added a citation (PMID: 1484548. Alpha-tubulin II is a male-specific protein in *Plasmodium falciparum*) which had reported the male-specific expression of alpha-tubulin II in the *Plasmodium* gametocytes.

In our study, the antibody we used is to specifically recognize the alpha-tubulin II. The information of the antibody was provided in the Material and Method.

Reviewer #2 (Remarks to the Author):

This manuscript by Yang et al describes work investigating the role of the microtubule-binding protein, EB1, in male gametogenesis in *Plasmodium yoelii*.

The authors show that PyEB1 is not expressed in asexual blood stage parasites and that knockout of PyEB1 did not affect asexual blood stage growth of gametocytogenesis.

PyEB1 is present from the initiation of gametocytogenesis, but more abundant in male gametocytes. Immunofluorescence microscopy, including Expansion Microscopy, and live cell imaging of EB1-GFP transfectants, following activation of male gametocytes yielded images consistent with EB1 association with the spindle.

Fig S2B. The diagram shows the SAS4 (basal body protein) as located inside the nucleus. Presumably this should always be in the cytoplasm.

Response: Thank reviewer for pointing this out. We have changed accordingly in Fig S4B.

Knockout of PyEB1 has no effect on female gamete formation; but leads to a decrease in the number of exflagellation centres. The failure to form midgut oocysts is therefore due to a male gamete defect. The defect is associated with failure to correctly segregate the DNA. The defect could be complemented with either PyEB1 or PfEB1, suggesting functional conservation.

In a very nice heterologous expression experiment, the authors transfected GFP-tagged PyEB1 and HsEB1 into MRC5 cells. They found that HsEB1 bound to the ends of spindle microtubules as expected, while PyEB1 bound along the MTs. They also purified recombinant GFP-tagged PyEB1 and HsEB1 and showed that PyEB1 seeds MT lattices and has MT stabilising activity, while HsEB1 tracks the plus end. The authors dissected the regions responsible for binding and found that both the CH domain and the linker have MT binding ability. They found that the N-terminal region is not required for MT binding but does play a role in chromosome

segregation.

Immunoprecipitation of EB1 revealed a potential interaction with NDC80 in activated male gametocytes. Given that EB1 is bound to microtubules, the immunoprecipitation protocol is likely to bring down microtubule fragments and a number of associated proteins. Thus, immunoprecipitation does not provide sufficient evidence to assume a direct interaction. However, I note that the authors are careful to conclude that “..EB1 regulates spindle-kinetochore attachment..” rather than concluding a direct interaction.

The authors have generated a very nice conditional knockdown system to look at the effects of NDC80 in male gametes. Not surprisingly, the data suggest that NDC80 is needed for kinetochore attachment.

The authors have also shown a critical role for EB1 Ser-15 phosphorylation for EB1 function in recruiting kinetochores to the spindle.

In the first part of the Discussion, the authors present the finding of PyEB1 binding along the length of the MTs as novel and surprising. They later point to recent studies in Pf (Ref 31) and *Toxoplasma gondii* (doi: 10.1091/mbc.E15-06-0437) showing the EB1 binds along the spindle MTs. Yeast EB1 has also been shown to bind along the MT lattice (DOI: 10.1038/ncomms11665). It might be better to nuance the Discussion to make it clear that the difference is with mammalian EB1, while the apicomplexan mitotic apparatus is more similar to that of yeast.

Response: We went to the publication (DOI: 10.1038/ncomms11665) mentioned by reviewer. In this work, they only investigated the human EB1 and concluded that the human EB1 binds the MT plus-end as expected.

Furthermore, several previous studies (PMID: 19667128 and 27872152) had shown that both the budding yeast EB1 homologue Bim1 and the fission yeast EB1 homologue Mal3 autonomously track the growing MT ends *in vitro*. See the below images from these publication (PMID: 19667128 and 27872152).

In conclusion, this is a very well-executed study that provides interesting insights into a potential role for EB1 in controlling spindle assembly and kinetochore attachment during male gametogenesis.

Response: We thank the reviewer for the encouraging comments on our work.

Minor points

Line 19. “Plasmodium evolves the endomitosis for this multinucleated cell division”. It may be better to say “Plasmodium undergoes endomitosis for this multinucleated cell division.” Endomitosis is a feature of all apicomplexans; and other organisms, such as yeast.

Response: We have changed “*Plasmodium* evolves the endomitosis for this multinucleated cell division” to “*Plasmodium* undergoes endomitosis in this multinucleated cell division” in line 19.

Line 23. Plasmodium EB1 is a unique orthologue. It is preferable not to use the term unique. As presented in Fig S11, Apicomplexan parasites likely have a functionally well-conserved EB1.

Response: We have changed “*Plasmodium* EB1 is a unique orthologue” to “the *Plasmodium* EB1 is an orthologue” in line 23.

Line 61. “A bipartite microtubule (MT) organization center (MTOC) [9], embedded in the nuclear envelope...”. While it is bipartite, only one part is embedded in the nuclear envelope.

Response: We have changed “A bipartite microtubule (MT) organization center (MTOC), embedded in the nuclear envelope, orchestrates.....” to “A bipartite microtubule (MT) organization center (MTOC), which associates with the nuclear envelope, orchestrates.....” in line 62.

Reviewer #3 (Remarks to the Author):

The study addresses the temporal stage-specific expression, localization and function of the EB1 protein in *Plasmodium yoelii*. The authors demonstrate that EB1 is dispensable for asexual and pre-sexual stages, but required for finalizing sexual differentiation and oocyst formation, a critical step in malaria transmission. The authors show, by complementation experiments, that restituting EB1, of either *P. yoelii* or *P. falciparum*, is sufficient for parasites to complete their life cycle in mosquitoes. They further analyze the basis of the phenotype, and decipher that EB1 is required for male gamete flagellation. The authors specifically identify that genome segregation is impaired during male gamete formation, while other male gamete structure assemble correctly. Furthermore, using a heterologous expression system the authors identify that *Plasmodium*'s EB1 does not accumulate at MT's plus-end (as does its human counterpart), but rather binds MTs evenly, deciphering a fundamental mechanistic difference between the two proteins. Finally, the authors identify that spindle MT-binding is given by EB1's CH and linker domains. The authors also map a specific phosphorylation required for EB1's localization to the spindle. The authors unravel that the MT lattice binding capacity of EB1 is encompassed by a distinct distribution of kinetochore proteins along spindle MTs, with EB1 playing a critical role in kinetochore clustering.

I would like to commend the authors for the truly spectacular images, for their remarkable attention to detail, and for thoroughly exploring multiple plausible hypotheses throughout the manuscript using an impressive array of approaches. The quality of the manuscript is very good, and overall, the writing is clear and concise. However, there are a few points that need addressing before the manuscript is in its final form for publication:

Response: We thank the reviewer for the encouraging comments on our work.

Line 103-104: the authors mention that “we reported that *Plasmodium* EB1 is a unique orthologue distinct from the eukaryotic EB1.” What do the authors mean? *Plasmodium* is an eukaryote, hence *Plasmodium*'s EB1 is by definition an eukaryotic EB1. Did they mean human EB1? A similar statement is repeated in Lines 249-250

Response:

In line 102-103, we have changed “In this study, we reported that *Plasmodium* EB1 is a unique orthologue distinct from the eukaryotic EB1” to “In this study, we report that the *Plasmodium* EB1 is an orthologue distinct from the canonical EB1”.

In line 246-247, we have changed “*Plasmodium* EB1 binds the full-length spindle MTs, suggesting MT-lattice affinity, which is distinct from the MT plus-end accumulation for the canonical EB1 in the eukaryotic cells” to “The *Plasmodium* EB1 binds the full-length spindle MTs, suggesting MT-lattice binding affinity, which is distinct from the MT plus-end binding for the canonical EB1”.

In Figure 1B a western blot is shown in which asexual blood stages and sexual stages are labeled. However, there are no markers for the stages that could confirm the majority of the proteome assayed for EB1 expression indeed corresponds to the indicated stages.

Response:

For stage expression analysis of EB1, we purified asexual blood stages and gametocyte stages of parasite for immunoblot. Under the microscopy after Giemsa staining, the asexual blood stages (including the ring, trophozoite, and schizont) and gametocyte (including the male and female) could be easily differentiated based on the cell stain characters (see the below picture, not included in the revised manuscript).

Line 128. It is stated that EB1 is more highly expressed in male gametocytes. The evidence provided for this statement is a single image in Figure 1C, which appreciably shows a higher fluorescence intensity signal for EB1 staining. However, there is no quantification of the signal, nor additional images for the reader to appreciate whether the images shown are representative of the variability between gametocyte types.

Response:

In Figure 1C, we displayed a representative image for EB1 expression in male and female gametocytes, respectively.

To address the reviewer's concern, we provided a raw image of gametocytes after EB1 staining (see the below picture, not included in the revised manuscript). Male gametocytes showed the cytosolic distribution of alpha-Tubulin II while female gametocytes were negative. EB1 was observed at the nucleus of both gender gametocytes with significantly higher expression level in male.

Figures 1L and 1J show that the \square EB1 strain generates oocysts in 7-10% of mosquitoes. This is neither mentioned nor the implication discussed in the manuscript. Are there alternative or redundant mechanisms? Could this be an indication of parthenogenesis? Are the oocysts formed viable (i.e. do they contain viable sporozoites)? Please address this point.

Response:

To evaluate the roles of EB1 in parasite development in the mosquito, $\Delta eb1$ parasite-infected mice were fed to the *Anopheles stephensi* mosquitoes. Midgut oocysts and salivary gland sporozoites were counted in the mosquitoes on day 7 and 14 post infection respectively. The mosquito infection prevalence is calculated as the ratio of the number of mosquitoes containing oocyst to the number of mosquitoes dissected. Compared to WT parasite (93.3%), the $\Delta eb1$ displayed much lower mosquito infection prevalence (6.8%) (Figure 1H). Notably, the mean value of oocyst numbers per mosquito are 85.6 in WT and 0.1 in $\Delta eb1$ respectively, indicating much less oocyst formation in the $\Delta eb1$ infection-positive mosquitoes. In addition, the $\Delta eb1$ oocysts failed to develop sporozoites in the mosquitoes (Figure 1I). The mosquito infection experiments were repeated independently twice.

In line 140-141, we have changed “The $\Delta eb1$ parasites produced no midgut oocyst or salivary gland sporozoite (Figure 1H and I)” to “The $\Delta eb1$ produced significantly less number of midgut oocysts (Figure 1H) and no salivary gland sporozoites (Figure 1I)”.

Figure 1J shows that the Comp1 strain expresses a higher level of EB1 despite it being under its endogenous promoter. Please address the origin of this difference in your results’ description, and how this could account for the slightly better complementation of the WT phenotype for Comp1 vs. Comp2 (results shown in Fig 1L). Could there be an effect of the expression level? Is *P. yoelii* better complemented by the *P. yoelii* protein than by the *P. falciparum* one? Please address this point

Response:

Both *P. yoelii* EB1 (PyEB1) in the Comp1 and *P. falciparum* EB1 (PfeB1) in the Comp2 were expressed under the endogenous promoter. In the Figure 1J, we observed

different immunoblot signals for PyEB1 and PfEB1. Two possible reasons may cause the difference. 1. The *P. falciparum* genome has an exceptionally high AT content compared to other *Plasmodium* species and eukaryotes in general –more than 80% in coding regions (PMID: 27994033). The difference in the codon usage between PyEB1 and PfEB1 may cause the different translation efficiency and protein production. 2. We collected the gametocyte samples for detecting PyEB1 and PfEB1 via immunoblot. Among the parasite samples after purification, the purity of gametocytes ranged from 50% to 90%. This may also cause the different immunoblot signals for PyEB1 and PfEB1.

We would not prefer to say that PyEB1 had better complementation than PfEB1. Because there is no significant difference in the oocyst number between *Comp1* and *Comp2*. See the p-value (red color) in the below picture (not included in the revised manuscript).

In general, there is reference in the figure legends to how many experiments were done per quantification. However, the sample size analyzed in each case is not mentioned. For example, In Figures 1F and G, the number of animals used in the quantifications is not specified. For figure 2A, how many individuals did you quantify per experimental replicate?

Response: We have added the information accordingly in the revised figure legend of Figures 1F and 1G, and Figure 2A.

For the results shown in Figure 1G, please provide the p-values obtained for both females and male gametes between experimental conditions.

Response: We have added the p-values information accordingly in the revised Figures 1G.

Line 179. The authors report that 67% of the WT gametes are nucleated. Please comment on whether obtaining a 33% anucleate male gametes is within the expected range and provide a citation.

Response:

Professor Robert Sinden had previously reported the anucleated male gamete in the *P. falciparum* (PMID: 27809). In the personal communication with Professor Mathieu

Brochet (University of Geneva), he also observed the anucleated male gamete (about 10-20%) in the *P. berghei*. Therefore it is a common phenomenon although the percentage of anucleated gamete varies between the *Plasmodium* species and experiments.

In the section of Discussion, we discussed this phenomena of anucleated male gametes.

In figure 3C, anti-tubulin labels multiple structures. It would be helpful for the non-expert reader to point out the mitotic spindle clearly in these images, or to provide insets where the overlap between the tubulin signal at the spindle and EB1 can be more clearly appreciated.

Response: Thank for reviewer's suggestion. We had added some white arrows indicating the mitotic spindle in the revised Figure 3C.

While it is well appreciated that the authors were thorough in showing that the localization of EB1 during gametogenesis is reproducible despite the tag or antibody used, I envision some of the redundant panels which serve as validation for the different strains could go onto supplementary figures. On the other hand, Figure 4S contains important topological information about EB1's with respect to other structures during male gametogenesis which are important to comprehend its function. This is important information which in my opinion should be part of a main figure.

Response:

Using both expansion microscopy and the EB1 as a spindle marker, we investigated relative position of the spindle to the adjacent structures including the nuclear envelope, nuclear pore, and basal body during the male gametogenesis. Basically, the results in our study (Figure S4) fit well with a recent study from the lab of Professor Mathieu Brochet(PMID:35077503), which had also showed the bipartite microtubule organization centre coordinating mitosis with axoneme assembly during male gametogenesis of *P. berghei*. Considering this, we prefer to put these results in the supplementary (Figure S4).

Minor suggestions:

The use of the English language should be thoroughly revised to ensure the reader's correct interpretation of the manuscript.

Response: We have a professional check the grammar and language throughout the manuscript.

To point out a few examples (please note this is definitely not a comprehensive list):

1. Plasmodium is frequently referred to as "the Plasmodium" while every mention to the mitotic spindle is missing the required article "the."

Response: Thank for reviewer pointing these out. We had changed accordingly.

2. There is a mix of verbal tenses used throughout the manuscript that should be revised. Please refer to your current findings, reported for the first time in this study, in the present tense (For example, in lines 103-104)

Response: Thank for reviewer pointing these out. We had changed accordingly.

3. Please correct all “Plasmodium EB1” to “Plasmodium’s EB1”

Response:

For reference, we read several *Plasmodium* parasite papers (PMID: 36030238, 36384964 and 36898988) published in Nature Communications and found that it is more common to use “*Plasmodium* protein” than “*Plasmodium*’s protein”. Therefore, we prefer to keep “The *Plasmodium* EB1” in the manuscript.

4. Either refer to “ the □EB1 strain” or simply refer to it as “□EB1”. Likewise, please refer to the EB1 protein, as either “the EB1 protein” or simply “EB1” (not “the EB1” alone).

Response: Thank for reviewer pointing these out. In the revised manuscript, we used “*Δeb1*”, “the *Δeb1* parasite”, or “the *Δeb1* strain”.

5. Lines 206-207. “EB1 was dispersed at nucleoplasm while Tubulin was dominantly at cytoplasm” please rephrase for clarity

Response: We changed “EB1 was dispersed at nucleoplasm while Tubulin was dominantly at cytoplasm” to “EB1 was dispersed in the nucleoplasm while Tubulin was mainly distributed in the cytoplasm” in line 206.

6. Lines 215-216, please rid of “the” in “we performed the Ultrastructure Expansion Microscopy,” and add “the” to “... fine localization of EB1 at the spindle”

Response: Thank for reviewer pointing these out. We had changed accordingly.

Line 109. “the spindle-kinetochore attachment was impaired for chromosome segregation in the male gametogenesis” ?? This sentence is unclear. Please rephrase for clarity.

Response: We changed “After genetic disruption of EB1, the spindle-kinetochore attachment was impaired for chromosome segregation in the male gametogenesis” to “Genetic disruption of EB1 leads to impaired spindle-kinetochore attachment for chromosome segregation in the male gametogenesis” in line 107.

Line 127. Please provide a reference for □-Tubulin as a marker for male gametocytes

Response:

In the revised manuscript, we added a citation (PMID: 1484548. Alpha-tubulin II is a

male-specific protein in *Plasmodium falciparum*) which had reported the male-specific expression of alpha-tubulin II in the *Plasmodium*.

Line 140. Please spell out *Anopheles stephensi*. To abbreviate, please refer to the species using appropriate nomenclature (*A. stephensi*)

Response: We changed “the *An. stephensi*” to “the *Anopheles stephensi*” at line 138.

Line 149: please replace “was” with “were”

Response: Thank for reviewer pointing it out. We had changed accordingly.

Line 540: please add “set” or “complement” between “haploid” and “of chromosomes”.

Response: Thank for reviewer’s suggestion. We changed “one haploid of chromosomes” to “one haploid set of chromosomes” in line 534.

Reviewer comments, second round review

Reviewer #1 (Remarks to the Author):

The resubmission has addressed all of the major issues except for one (see below).

As noted previously, the manuscript by Yang et al. investigates the role of EB1 during male gamete formation in Plasmodium parasites. The connections between the spindle and kinetochores are essential and have not been extensively studied in Plasmodium. The authors demonstrate that EB1 has unexpected features, including loss of plus-end binding to microtubules. EB1 association with microtubules instead occurs throughout the "microtubule lattice". This manuscript utilizes the rodent parasite, Plasmodium yoelii, as the major model. The authors use high-quality genetics for both knockout and complementation studies, adding to the robustness of the results.

The study provides an interesting and in-depth analysis of EB1. The findings will be important for the parasitology and larger cell biology communities.

OUTSTANDING ISSUE:

In the initial review, I asked about the specificity of the anti-tubulin II antibody (Sigma T6199). The authors use this antibody to identify male gametocytes, because alpha tubulin II is (mostly) male specific. In my first review, I was not clear enough about my request. The epitope recognized by the antibody that is used is listed on the Sigma website as amino acids 426-430 of the 488 amino acid alpha tubulin. This epitope, AALEK, is present in both alpha tubulin I and alpha tubulin II -- in Py, Pf, and many other organisms. Therefore, the antibody is not specific. Rather, it likely just sees the most abundant form of tubulin in gametocytes, and female gametocytes do not have much tubulin (I or II). The use of this antibody should NOT be said to be specific for type II unless the authors can provide a proper citation for this particular antibody being specific for type II alpha tubulin. The text and the methods need to be corrected.

Reviewer #2 (Remarks to the Author):

The authors have successfully addressed my suggestions.

Reviewer #3 (Remarks to the Author):

I thank the authors for thoroughly addressing my concerns in the revised version of the manuscript. I have no further observations.

Response to Reviewer Comments on the manuscript [NCOMMS- 23-04049]:

Reviewer #1

The resubmission has addressed all of the major issues except for one (see below).

As noted previously, the manuscript by Yang et al. investigates the role of EB1 during male gamete formation in Plasmodium parasites. The connections between the spindle and kinetochores are essential and have not been extensively studied in Plasmodium. The authors demonstrate that EB1 has unexpected features, including loss of plus-end binding to microtubules. EB1 association with microtubules instead occurs throughout the “microtubule lattice”. This manuscript utilizes the rodent parasite, Plasmodium yoelii, as the major model. The authors use high-quality genetics for both knockout and complementation studies, adding to the robustness of the results.

The study provides an interesting and in-depth analysis of EB1. The findings will be important for the parasitology and larger cell biology communities.

OUTSTANDING ISSUE:

In the initial review, I asked about the specificity of the anti-tubulin II antibody (Sigma T6199). The authors use this antibody to identify male gametocytes, because alpha tubulin II is (mostly) male specific. In my first review, I was not clear enough about my request. The epitope recognized by the antibody that is used is listed on the Sigma website as amino acids 426-430 of the 488 amino acid alpha tubulin. This epitope, AALEK, is present in both alpha tubulin I and alpha tubulin II -- in Py, Pf, and many other organisms. Therefore, the antibody is not specific. Rather, it likely just sees the most abundant form of tubulin in gametocytes, and female gametocytes do not have much tubulin (I or II). The use of this antibody should NOT be said to be specific for type II unless the authors can provide a proper citation for this particular antibody being specific for type II alpha tubulin. The text and the methods need to be corrected.

Response: Thank reviewer for pointing it out. We have changed the expression in both text, the method section, and figure legend.

In line 125, we changed “the gametocytes were stained with antibodies against \$\alpha\$ -Tubulin (a male gametocyte marker) and HA.” to “the gametocytes were stained with the antibody recognizing both \$\alpha\$ -Tubulin I and \$\alpha\$ -Tubulin II proteins (a male gametocyte marker) as well as anti-HA antibody.”

In line 701, we changed “Parasites were then incubated with anti- \$\alpha\$ -Tubulin II antibody (Sigma-Aldrich, cat#T6199)” to “Parasites were then incubated with an antibody

(Sigma-Aldrich, cat#T6199) recognizing both α -Tubulin I and α -Tubulin II proteins”.

In line 714, we changed “Parasites were then incubated with anti- α -Tubulin II antibody (Sigma-Aldrich, cat#T6199)” to “Parasites were then incubated with an antibody (Sigma-Aldrich, cat#T6199) recognizing both α -Tubulin I and α -Tubulin II proteins”.

In line 728, we changed “mouse anti- α -Tubulin II” to “mouse antibody against both α -Tubulin I and α -Tubulin II”.

In legend of Figure 1c, we changed “Co-staining the *eb1::6HA* gametocytes with antibodies against HA and α -Tubulin II (male gametocyte marker)” to “Co-staining the *eb1::6HA* gametocytes with anti-HA antibody and antibody against both α -Tubulin I and α -Tubulin II (male gametocyte marker)”.

Reviewer #2

The authors have successfully addressed my suggestions.

Reviewer #3

I thank the authors for thoroughly addressing my concerns in the revised version of the manuscript. I have no further observations.